# ABA-mediated regulation of rice grain quality and seed dormancy via the NF-YB1-SLRL2-bHLH144 Module

Jin-Dong Wang[1,3], Jing Wang[1,3], Li-Chun Huang[1], Li-Jun Kan[1], Chu-Xin Wang[1], Min Xiong[1], Peng Zhou[1], Li-Hui Zhou[1], Chen Chen [1], Dong-Sheng Zhao[1], Xiao-Lei Fan[1], Chang-Quan Zhang[1], Yong Zhou[1], Lin Zhang[1], Qiao-Quan Liu [1,2] ✉ & Qian-Feng Li [1,2] ✉

Abscisic acid (ABA) plays a crucial role in promoting plant stress resistance and seed dormancy. However, how ABA regulates rice quality remains unclear. This study identifies a key transcription factor SLR1-like2 (SLRL2), which mediates the ABA-regulated amylose content (AC) of rice. Mechanistically, SLRL2 interacts with NF-YB1 to co-regulate *Wx*, a determinant of AC and rice quality. In contrast to SLR1, SLRL2 is ABA inducible but insensitive to GA. In addition, SLRL2 exhibits DNA-binding activity and directly regulates the expression of *Wx*, *bHLH144* and *MFT2*. SLRL2 competes with NF-YC12 for interaction with NF-YB1. NF-YB1 also directly represses *SLRL2* transcription. Genetic validation supports that *SLRL2* functions downstream of *NF-YB1* and *bHLH144* in regulating rice AC. Thus, an NF-YB1-SLRL2-bHLH144 regulatory module is successfully revealed. Furthermore, SLRL2 regulates rice dormancy by modulating the expression of *MFT2*. In conclusion, this study revealed an ABA-responsive regulatory cascade that functions in both rice quality and seed dormancy.

Rice is the staple food for over half of the global population. Breeding elite rice varieties with high yields and superior quality is crucial for preventing hunger and improving people's quality of life, especially under deteriorated environmental conditions. Although impressive achievements have been got in rice yield, the progress of rice quality improvement lags far behind. This may be mainly caused by the complexity of rice quality research and the different preferences of different consumers[1]. In a broader context, excellent seed-related traits, including seed size, grain composition, and seed germination and dormancy characteristics influence both yield and desired quality of rice. Starch is the major component of rice, and its composition and physicochemical properties determine rice quality, especially eating and cooking quality (ECQ). Currently, amylose content (AC), gel

consistency (GC), gelatinization temperature (GT), and the viscosity of rice are still the most widely accepted indicators for proper evaluation and estimation of rice ECQ[2]. All these physicochemical parameters are intricately linked to starch properties, where AC is the most critical parameter determining rice ECQ. Unlike amylopectin synthesis, which is controlled by more than ten different enzymes, amylose biosynthesis is catalyzed by a single enzyme, granule-bound starch synthase I (GBSSI), encoded by the *Wx* gene. Notably, the *Wx* gene is a major gene in AC and GC regulation, and also a minor contributor to rice GT. Therefore, the *Wx* gene is considered to be the most important quality-determining gene in rice quality improvement. Recently, extensive efforts have been directed toward identifying and applying the beneficial natural *Wx* alleles which can mildly modulate AC and

[1]Jiangsu Key Laboratory of Crop Genomics and Molecular Breeding/Zhongshan Biological Breeding Laboratory/ Key Laboratory of Plant Functional Genomics of the Ministry of Education, College of Agriculture, Yangzhou University, Yangzhou 225009 Jiangsu, China. [2]Co-Innovation Center for Modern Production Technology of Grain Crops of Jiangsu Province, Yangzhou University, Yangzhou 225009 Jiangsu, China. [3]These authors contributed equally: Jin-Dong Wang, Jing Wang. ✉e-mail: qqliu@yzu.edu.cn; qfli@yzu.edu.cn

consequently improve rice ECQ in breeding strategies[3,4]. For example, the soft rice, which is known for its good taste and high ECQ, contains the $Wx^{mp}$ or $Wx^{mw}$ alleles. The AC of soft rice is about 12%, while that of traditional rice with the $Wx^b$ allele is about 15%[4]. In addition, the recent use of the CRISPR/Cas9 technique to modify the $Wx$ promoter has also generated a number of novel $Wx$ alleles, further enriching the elite $Wx$ allele pools for fine-tuning rice AC[5,6]. Thus, AC fine-tuning is a major contributor to the improvement of rice ECQ.

While the pathways responsible for starch biosynthesis are well understood, the molecular regulatory network of starch biosynthesis and the underlying recipe for producing high-quality rice are still largely elusive. Recent strides have been made in identifying several transcription factors have been reported to directly regulate the expression of $Wx$ gene[7–11]. Notably, NF-YB1, a rice endosperm-specific core transcription factor, can directly bind to the $Wx$ gene promoter, regulating its expression[8]. In the pursuit of enhanced rice yield and quality, some other seed-related traits are also critical, especially in the context of deteriorating environmental conditions. For example, an appropriate dormancy trait is critical to prevent pre-harvest sprouting (PHS) during the seed maturation stage, especially under hot and humid conditions. PHS in crops, including rice, has posed a challenge worldwide, contributing not only to yield losses but also diminished grain quality and vitality[12–15].

Plant hormones have profound physiological effects on plant biology, orchestrating plant-intrinsic developmental events as well as mediating environmental inputs[4,16]. Abscisic acid (ABA) is a classical phytohormone that plays a central role in plant abiotic stress resistance and also has versatile functions in promoting seed dormancy and stomatal closure, suppressing seed germination and root growth[17]. In addition, ABA is also a positive regulator of rice grain filling by modulating starch biosynthesis[18]. The physiological evidence indicated that ABA treatment could slightly reduce rice AC, which has potential application in improving rice grain quality[19]. $NCED$ genes control the limiting step of ABA biosynthesis in plants. For example, $OsNCED3$ is highly expressed in developing seeds and is involved in the regulation of rice grain development and PHS[20]. In addition, the $DG1$ gene controls the transport of ABA from rice stems to seeds, and the filling rate of $dg1$ mutant seeds was slower[21]. Although these evidences indicated that ABA is involved in the regulation of grain filling and rice quality, the precise mechanism by which ABA regulates rice quality has not been reported. Here, we successfully identified a transcriptional regulator SLR1-like2 (SLRL2), and demonstrated that it is the key gene mediating the regulation of ABA on rice quality. Consistent with the suppressive effect of ABA on AC, overexpression of $SLRL2$ also reduced rice AC. Remarkably, in contrast to SLR1, SLRL2 functions as a canonical transcription factor with DNA-binding activity, thereby directly regulating the expression of both $Wx$ and $bHLH144$. At the same time, SLRL2 also competes with NF-YC12 for NF-YB1 and reduces the stabilizing effect of NF-YC12-bHLH144 on NF-YB1. Furthermore, NF-YB1 can also directly regulate the transcription of $SLRL2$. Thus, this study successfully uncovers an ABA-responsive molecular mechanism central to rice quality regulation, centered on the NF-YB1-SLRL2-bHLH144 regulatory module, with the $Wx$ gene as its main target. Such fine regulation of rice AC is helpful in achieving a balance between rice palatability and rice yield. In addition, SLRL2, bHLH144, and NF-YB1 are also involved in the regulation of rice PHS. Significantly, SLRL2 regulates rice PHS by directly targeting a key gene, $MFT2$, making the $SLRL2$ gene useful for generating elite rice varieties with multiple improved seed traits.

## Results

### Identification of candidate transcription factors mediating ABA regulation of rice AC

We treated rice panicles at the filling stage with ABA or fluridone (a carotenoid and ABA biosynthesis inhibitor), which resulted in a reduced 1000-grain weight of harvested rice seeds mainly due to the reduced grain thickness (Supplementary Fig. 1a–d). Notably, ABA treatment significantly decreased rice AC and increased rice GC, whereas fluridone treatment showed an opposite effect on AC but no significant effect on GC (Fig. 1a and Supplementary Fig. 1e). To further confirm the relationship between ABA and rice quality, especially rice AC, which has not yet been mechanically dissected yet, $nced2$ and $dg1$ mutants, which are involved in ABA biosynthesis and transport, respectively, were generated (Supplementary Fig. 1f, g). Grain quality analysis showed that the mutation of either $NCED2$ or $DG1$ significantly increased rice AC and decreased GC (Fig. 1b and Supplementary Fig. 1h), which was consistent with the results of fluridone treatment. Therefore, both pharmacological experiments and rice mutant analyses demonstrated that ABA is involved in the regulation of rice quality by modulating AC and GC.

To elucidate the underlying molecular mechanisms, we postulated that ABA might modulate the expression of certain transcription factors, which, in turn, regulate key genes that influence grain quality, such as the $Wx$ gene, which is responsible for amylose synthesis. To verify our hypothesis, we first examined the expression of the $Wx$ gene in response to ABA and fluridone treatment. The results showed that ABA decreased $Wx$ expression while fluridone increased it (Fig. 1c and Supplementary Fig. 2a). The result of $Wx$ expression analysis in the $nced2$ and $dg1$ mutants further confirmed this conclusion (Fig.1d and Supplementary Fig. 2b). Next, experiments using a series of ABA concentration gradients and different treatment times showed that $Wx$ expression was sensitively inhibited by ABA in a dose-independent manner (Fig.1e and Supplementary Fig. 2c). Considering the pivotal role of $Wx$ in determining rice AC and quality, and its expression regulated by ABA, we used three databases, including Rice FREND, RiceXPro (https://ricexpro.dna.affrc.go.jp/) and PlantTFDB (http://planttfdb.gao-lab.org/), to screen the candidate transcription factor encoding genes that should be sensitive to ABA and have a tight co-expression pattern with $Wx$ (Supplementary Data 1). From this analysis, two candidate genes, $SLRL2$ and $NF-YC9$, met the selection criteria (Supplementary Fig. 2d, e). Further experimental verification revealed that the expression of $SLRL2$ was strongly induced by ABA and suppressed by fluridone (Fig. 1f–h), while that of $NF-YC9$ was only slightly affected by ABA (Fig. 1f, g). Spatial-temporal expression pattern analysis revealed that only the $SLRL2$ gene shared a consistent expression pattern with $Wx$ in developing seeds (Supplementary Fig. 2f–h). In situ hybridization analysis revealed that $SLRL2$ was expressed in the pericarp, aleurone layer, and starchy endosperm (Supplementary Fig. 2i, j). In addition, the $SLRL2$ co-expressed genes were enriched in starch and sucrose metabolism pathways (Supplementary Fig. 2k). All these evidences suggested that $SLRL2$ should be the target transcription factor.

Phylogenetic tree analysis revealed that three rice proteins, including SLR1, SLRL1 (SLR1 like 1) and SLRL2 (SLR1 like 2), belonged to the DELLA subfamily (Supplementary Fig. 3a). In addition, all Arabidopsis DELLA proteins were also members of this family. These DELLA family proteins can be divided into two groups based on the presence or absence of the DELLA domain. For example, SLR1 was the canonical DELLA protein with a DELLA domain, whereas SLRL1 and SLRL2 were without the domain (Supplementary Fig. 3b). Expression analysis showed that $SLRL2$ was insensitive to GA at both transcript and protein levels (Supplementary Fig. 4a–c). Subcellular localization analyses revealed that SLRL2 was localized to the nucleus (Supplementary Fig. 4d). In addition, only the full-length SLRL2 had transcriptional activation activity (Supplementary Fig. 4e–h).

### Role of SLRL2 as a positive regulator of rice ECQ

To elucidate the functions of $SLRL2$ and its potential mediation of ABA's influence on rice AC, both the $SLRL2$ knockout mutant and overexpression transgenic rice were generated. First, two independent homozygous $slrl2$ mutant lines were obtained by using the CRISPR/

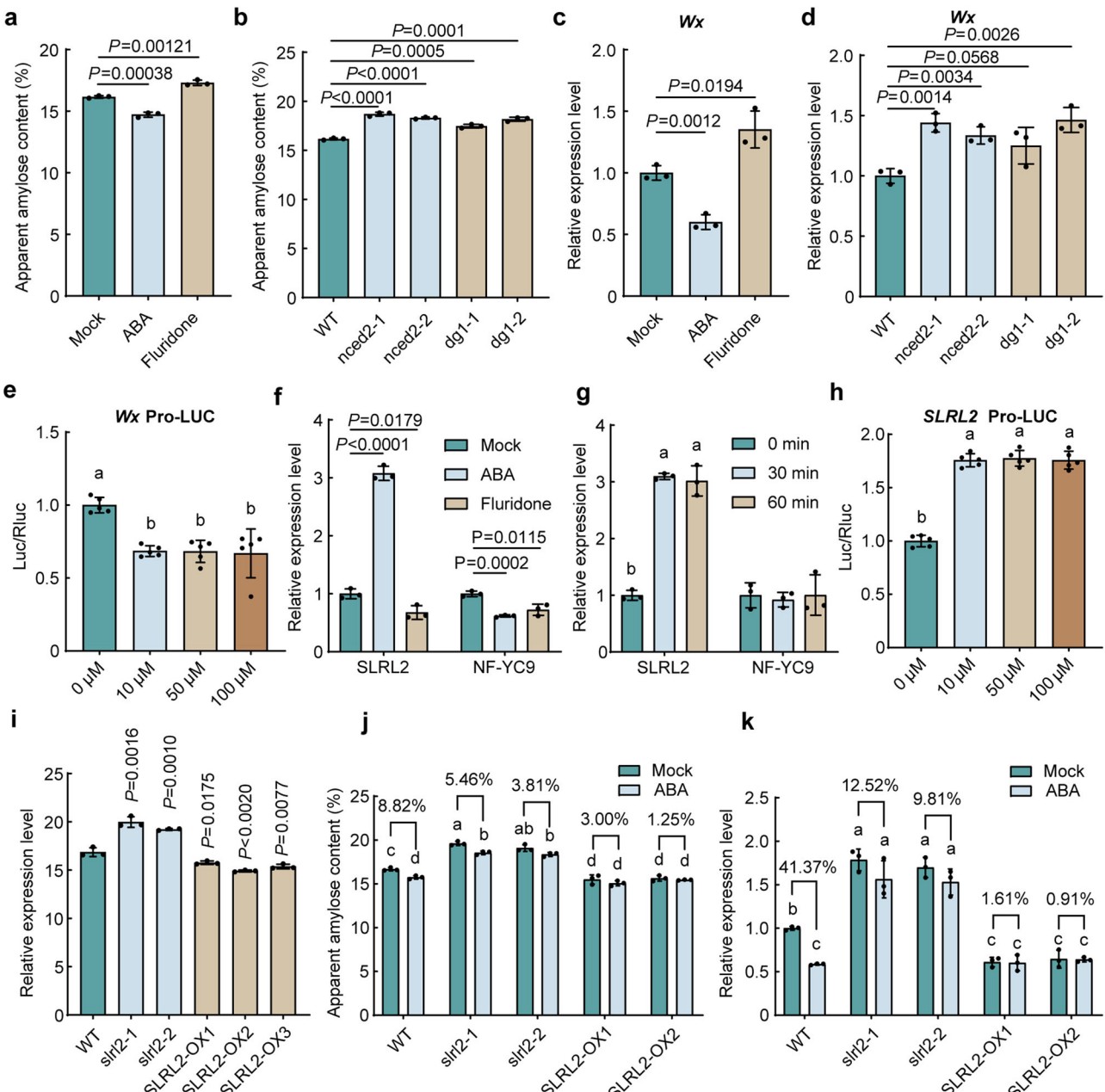

**Fig. 1 | SLRL2 is a candidate transcription factor mediating ABA regulation of amylose content (AC) in rice. a** Rice AC in response to ABA or fluridone treatment. **b** AC of *nced2* and *dg1* mutants. **c** Expression of *Wx* in response to ABA or fluridone treatment. 100 µM ABA or 100 µM fluridone was sprayed every two days on rice panicles at 5 days after flowering (DAF), and grain samples were collected at 15 DAF. **d** Expression of *Wx* in the mature seeds of *nced2* and *dg1* mutants. In (**a**–**d**), data are means ± SD (*n* = 3 biological replicates) and comparisons are made by two-tailed Student's *t* test. **e** Dual luciferase reporter assay to study the regulation of *Wx* gene expression to ABA treatment. Fluorescence ratio was measured after treatment with a series of ABA concentration gradients (0, 10, 50 and 100 µM) for 30 min. Error bars represent SD (*n* = 5 biological replicates). **f** Expression of *SLRL2* and *NF-YC9* in response to ABA or fluridone treatment. **g** Expression of *SLRL2* and *NF-YC9* under short-term ABA treatment. The 14-d-old rice seedlings were transferred to a medium containing 50 µM ABA, and rice samples were collected just before and after 30 and 60 min of ABA treatment, respectively. In (**c**, **d**, **f**, **g**), *OsUBC1* was used

as an internal control to normalize gene expression. The expression level of the target gene was set to 1 for samples without ABA treatment or in the wild-type control. In (**f**, **g**), error bars represent SD (*n* = 3 biological replicates). **h** Dual luciferase reporter assay was used to examine the response of *SLRL2* to different concentrations of ABA treatment. Error bars represent the SD (*n* = 5 biological replicates). **i** AC of the *SLRL2* knockout and overexpression transgenic plants and the WT control. **j** AC of *SLRL2* knockout and overexpression materials in response to ABA treatment. **k** *Wx* expression in *SLRL2* knockout and overexpression materials in response to ABA treatment. Data in e-k are means ± SD (*n* = 5) biological replicates in (**e**, **h**); *n* = 3 biological replicates in (**f**, **g**, **i**, **j**, **k**). A two-sided Student's paired *t* test was used to generate the *P* values in (**f**, **i**). In (**e**, **g**, **h**), different letters indicate significant differences (*P* < 0.05, one-way ANOVA with two-sided Tukey's HSD test). In (**j**, **k**), a two-way ANOVA with two-sided Tukey's HSD test was used to generate the *P* values, *p* < 0.05. *P* values are adjusted and shown in the Source Data file.

Cas9 genome editing system, designated as *slrl2-1* and *slrl2-2* (Supplementary Fig. 5a). Meanwhile, the *SLRL2*-overexpression lines were also successfully generated and three representative homozygous lines were selected for the subsequent analyses (Supplementary

Fig. 5b, c). Notably, we further demonstrated that SLRL2 was indeed insensitive to GA by analyzing the *SLRL2*-OX rice (Supplementary Fig. 5d). The phenotypic assay showed that the *SLRL2* mutation had no observable effects on the major agronomic traits of rice

(Supplementary Fig. 5e–i). However, *SLRL2* overexpression resulted in reduced plant height and flag leaf width (Supplementary Fig. 5e–i).

Since ABA treatment affected both grain shape and rice quality, we were particularly interested in traits associated with rice seeds. Grain shape analysis data showed that neither knockout nor overexpression of *SLRL2* affected grain shape and rice yield (Supplementary Fig. 5j–o), indicating that *SLRL2* responds only specifically to ABA-regulated rice quality. The results of rice quality analyses showed that the AC was increased in the *slrl2* mutants and decreased in the *SLRL2* overexpression lines (Fig. 1i), while their total starch content was not altered (Supplementary Fig. 6a). In line with the AC alterations, *Wx* expression also increased in the *slrl2* mutant but reduced in the *SLRL2* overexpression plants (Supplementary Fig. 6b, c). As expected, GC in these rice materials showed an opposite change to AC (Supplementary Fig. 6d). Therefore, the changes in AC and GC in *SLRL2* overexpression lines were consistent with those in ABA-treated rice and opposite to those in *nced2* and *dg1* mutants. Most importantly, *SLRL2* mutation attenuated the effect of ABA on rice AC, whereas *SLRL2* overexpression mimicked the effect of ABA (Fig. 1j), suggesting that SLRL2 should be the regulator mediating ABA in the regulation of rice quality. This notion is also supported by *Wx* expression analysis in ABA-treated *slrl2* mutants and *SLRL2* overexpression lines (Fig. 1k). Furthermore, modulation of *SLRL2* had no significant effect on rice GT (Supplementary Fig. 6e, f), but *SLRL2* mutation increased while its overexpression decreased the setback viscosity (SBV) (Supplementary Fig. 6g, h). In general, high GC, low AC and SBV were often positively correlated with a good ECQ. Indeed, the taste value data directly supported this notion, i.e., *SLRL2* overexpression improved the ECQ of rice while its mutation decreased it (Supplementary Fig. 6i). Therefore, these results suggest that *SLRL2* overexpression could improve rice quality.

## Direct interaction of SLRL2 with NF-YB1 in regulating *Wx* expression

While DELLA proteins typically lack direct DNA-binding activity, they often interact with other transcription factors to modulate downstream gene expression. We speculated that SLRL2 might also modulate *Wx* expression in this way. Therefore, yeast two-hybrid screening was performed to identify SLRL2-interacting transcription factors by using a rice seed cDNA library. Fortunately, NF-YB1, an important seed-specific transcription factor in rice, was identified (Supplementary Fig. 7a). Their interaction was then verified by a series of in vitro and in vivo experiments. First, the result of the yeast two-hybrid analysis confirmed the library screening result (Fig. 2a). The SLRL2 protein was further split into seven sections, labeled N1 to N4 and C1 to C3, based on its conserved functional domains (Supplementary Fig. 7b). Yeast two-hybrid analysis revealed that the PFYRE domain (section C3) of SLRL2 mediated its interaction with NF-YB1 (Fig. 2b and Supplementary Fig. 7c). Second, the pull-down assay further verified that NF-YB1-MYC protein specifically interacted with SLRL2-Flag protein, but not with bHLH144-Flag protein (Fig. 2c). This provides strong evidence that SLRL2 and NF-YB1 interact directly in vitro.

Next, three approaches were applied to demonstrate the SLRL2–NF-YB1 interaction in vivo. First, their interaction was verified in *Nicotiana benthamiana* leaf epidermis cells by using a split luciferase complementation assay (SLCA) (Fig. 2d). Second, a bimolecular fluorescence complementation (BiFC) assay confirmed that SLRL2 can also interact with NF-YB1 mainly in the nucleus (Fig. 2e). Finally, the Co-IP assay showed that the SLRL2-Flag protein can specifically co-precipitate the NF-YB1-MYC protein (Fig. 2f). Collectively, all the above data demonstrated the interaction between SLRL2 and NF-YB1 in rice.

## SLRL2 represses *Wx* expression via multiple pathways

The DELLA protein frequently employs the strategy of restricting the transcriptional activation activity of its interacting partner[22,23]. To evaluate whether SLRL2 also uses a similar approach to affect NF-YB1, a dual-luciferase reporter system was used. The analysis result showed that NF-YB1 had a strong transactivation activity, while co-expression of SLRL2 notably suppressed its activity (Fig. 3a and Supplementary Fig. 8a). The fact that NF-YB1 also prevented SLRL2 from activating transcription, however, indicates that SLRL2 and NF-YB1 are antagonistic to one another. Previous studies have demonstrated that NF-YB1 can directly bind to the G-box of the *Wx* gene promoter. Therefore, we tested whether SLRL2 affects NF-YB1 in its regulation of *Wx* expression (Supplementary Fig. 8a). The result showed that SLRL2 inhibited while NF-YB1 activated *Wx* transcription, and their co-expression led to an attenuated *Wx* expression (Fig. 3b). Next, we co-transformed a mutated SLRL2 with PFYRE domain deleted (SLRL2△C3) to study its effect on NF-YB1 and *Wx* promoter. Surprisingly, the natural SLRL2 protein and the SLRL2△C3 protein both had a similar inhibitory effect on *Wx* expression (Fig. 3b), suggesting a more complex regulatory mechanism may exist.

To further confirm that SLRL2 and NF-YB1 can form a transcriptional protein complex to co-regulate *Wx* transcription, the chromatin immunoprecipitation (CHIP) assay was performed. First, an anti-NF-YB1 antibody was used and the ChIP-qPCR data showed that the P1 region of the *Wx* promoter was significantly enriched by NF-YB1 (Fig. 3c and Supplementary Fig. 8b). Then, the potential in-vivo interaction between the SLRL2 and *Wx* promoter was also examined in the seeds of *SLRL2-Flag* transgenic rice. Consistent with the ChIP-qPCR result of NF-YB1, the P1 region of the *Wx* promoter was also notably enriched by SLRL2 (Fig. 3d), supporting that SLRL2 and NF-YB1 can form a protein complex to co-regulate *Wx* expression. Finally, the EMSA experiment was used to test whether SLRL2 modulates *Wx* expression by affecting the binding activity of NF-YB1 on the *Wx* promoter. As expected, NF-YB1 exhibited direct binding to the *Wx* promoter (Fig. 3e). Notably, the presence of native SLRL2 noticeably attenuated NF-YB1's binding efficiency. Intriguingly, deletion of the PFYRE domain of SLRL2 abolished this inhibitory effect (Fig. 3f), strongly implying that SLRL2 plays a negative regulatory role in *Wx* expression.

Given that protein stability also plays a key role in controlling NF-YB1 in rice, a cell-free degradation analysis was performed to test whether SLRL2 affects NF-YB1 stability. The result indicated that when only NF-YB1-MYC was introduced to the system, significant NF-YB1 degradation was observed after 15 min (Fig. 3g). When both NF-YB1-MYC and SLRL2-Flag proteins were co-incubated, the ratio of NF-YB1 degradation was significantly delayed. However, SLRL2△C3-Flag had no effect on NF-YB1 degradation rate (Fig. 4g). This observation affirms that the protective effect of SLRL2 on NF-YB1 was based on their direct protein-protein interaction. On the other hand, the SLRL2 or the SLRL2△C3 protein itself was very stable (Fig. 3g). Consequently, these results showed that SLRL2 interaction with NF-YB1 stabilizes the NF-YB1 protein.

Next, a mutant *Wx* promoter with the G-box (in the P1 region) removed, hence named *Wxpro△G-LUC*, was used to further investigate whether SLRL2 can regulate *Wx* expression independently of NF-YB1. Deletion of the G-box only attenuated but did not abolish, the activation effect of NF-YB1 on the *Wx* gene. Furthermore, the deletion did not affect the regulation of the *Wx* gene by SLRL2 (Fig. 3h). Importantly, the presence of both SLRL2 and NF-YB1 resulted in intermediate levels of *Wx* transcription. These data suggest that SLRL2 itself may be a transcriptional repressor of the *Wx* gene independent of NF-YB1. Data from the dual-luciferase reporter assay using the *nf-yb1* mutant further supported this notion (Fig. 3i). To exclude the possibility that SLRL2 interacts with other transcription factors with *Wx*-promoter binding activities to co-regulate *Wx* transcription[7,9,10], yeast two-hybrid experiments were performed and the results showed that SLRL2 did not interact with these reported transcription factors (Supplementary Fig. 8c). Although there is no direct evidence, these

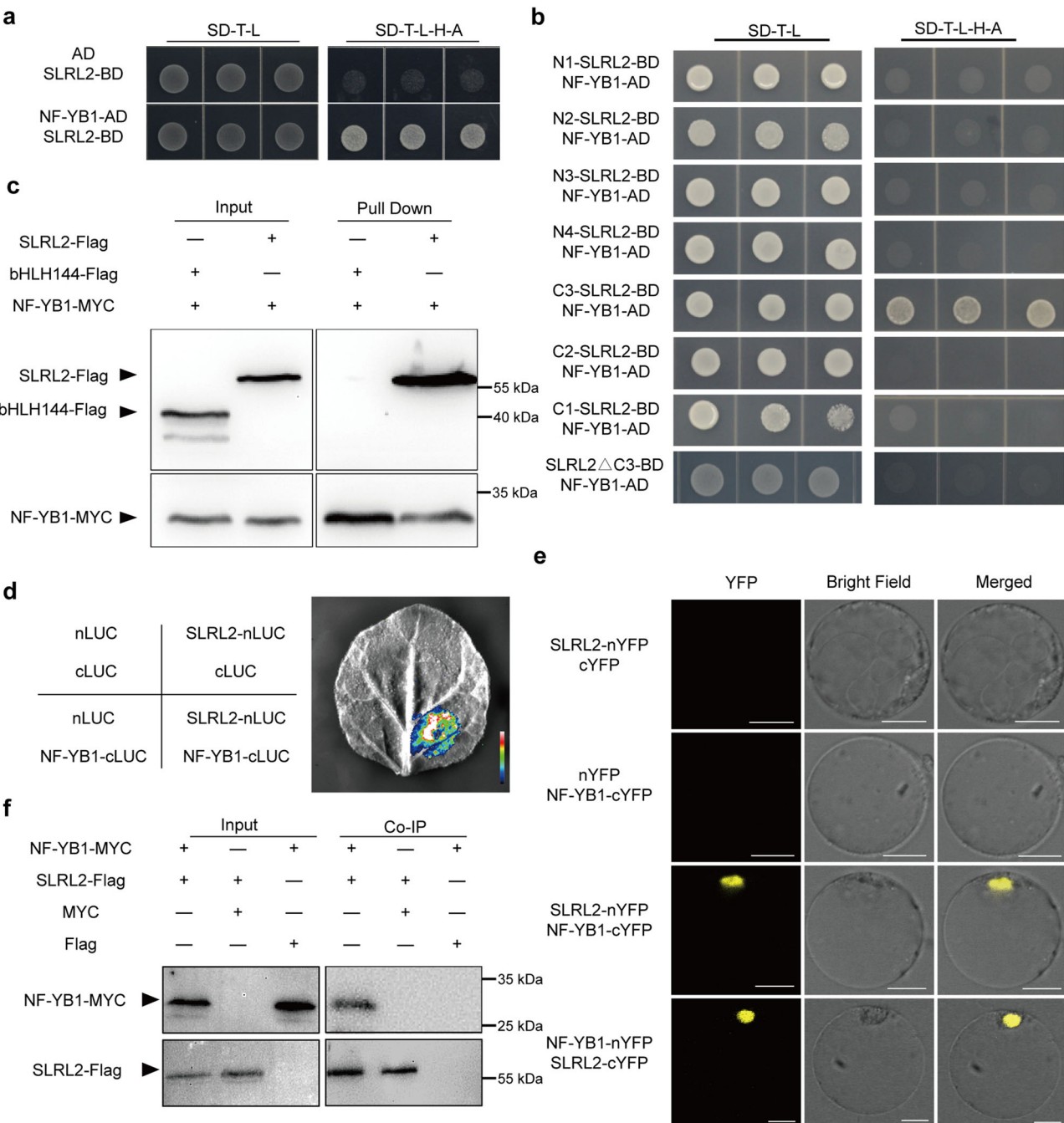

**Fig. 2 | Interaction between SLRL2 and NF-YB1. a** Yeast two-hybrid analysis verified the interaction between SLRL2 and NF-YB1. NF-YB1 was cloned into the AD vector and SLRL2 was cloned into the BD vector, respectively. Positive interactions were detected by using the SD dropout medium with 25 mM 3AT inhibitor. **b** Confirmation of the interaction domain of SLRL2 protein mediating its interaction with NF-YB1 by yeast two-hybrid analysis. **c** Pull-down experiment confirmed the interaction between SLRL2 and NF-YB1 in vitro. Purified recombinant MYC-NF-YB1 protein was used to pull down SLRL2-Flag protein by using amylose beads. Anti-MYC and anti-Flag antibodies were used for immunoblot detection. MYC-NF-YB1 and bHLH144-Flag protein groups were used as the negative controls. The experiment was independently repeated three times with similar results. **d** Split luciferase complementation assay (SLCA) demonstrated the in vivo interaction between SLRL2 and NF-YB1. Three sets of construct combinations, including nLUC and cLUC, SLRL2-nLUC and cLUC, nLUC, and NF-YB-cLUC, were transformed and used as negative controls. The fluorescence intensity in the image ranges from 2051 to 13144. **e** Bimolecular fluorescence complementation (BiFC) analysis revealed the nuclear interaction between SLRL2 and NF-YB1 in rice protoplasts. Two sets of plasmid groups, including SLRL2-nYFP and cYFP, nYFP and NF-YB1-cYFP, were used as negative controls. The scale bar is 10 uM. **f** Co-immunoprecipitation (Co-IP) assay further confirmed their interaction in vivo. In this experiment, three sets of construct combinations, including NF-YB1-MYC and SLRL2-Flag, MYC and SLRL2-Flag, and NF-YB1-MYC and Flag, were also co-transformed and transiently expressed in rice protoplasts. Total proteins were extracted and incubated with anti-Flag antibody conjugated to magnetic beads. The immunoprecipitated proteins were detected by Western blot using anti-Myc and anti-Flag antibodies, respectively. **c, f** Plus (+) and minus (-) signs indicate the presence or absence of the indicated protein in each group. The experiments in (**e, f**) have three biological replicates with similar results. The source data for (**c, e, f**) are provided in a Source Data file.

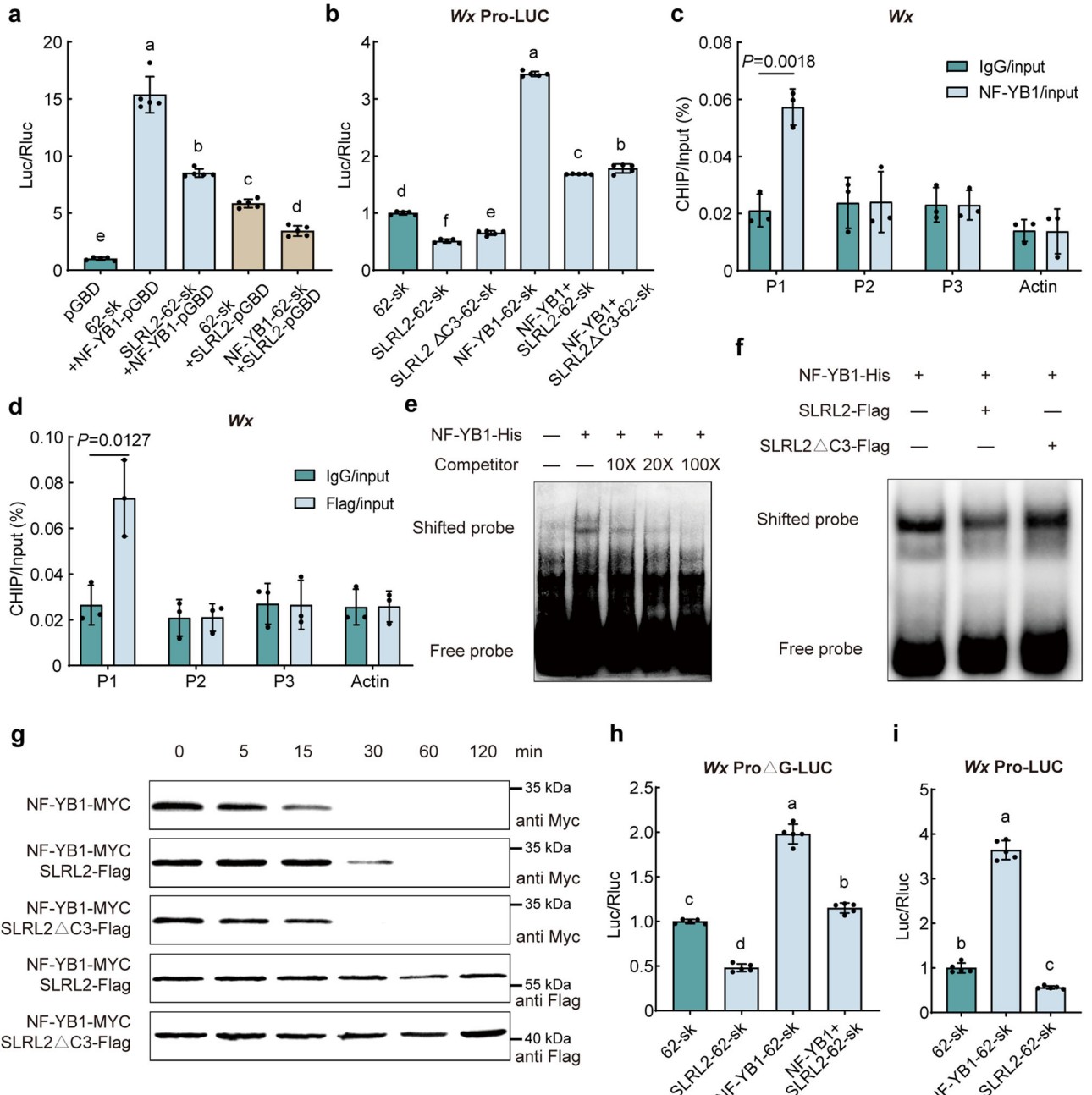

**Fig. 3 | SLRL2 and NF-YB1 regulation of *Wx* transcription. a** Transcriptional activity analysis of the interaction between SLRL2 and NF-YB1. The experimental value of the pGBD and 62-SK co-transformation group was set to 1. 62-SK is an empty vector for overexpression of the target gene under the control of the 35S promoter. **b** Dual-luciferase reporter assay to study the regulation of *Wx* gene expression by SLRL2 and NF-YB1. The value from the 62-SK and *Wx*pro-LUC co-transformation group was set to 1. In (**a**, **b**), data are means ± SD (*n* = 5 biological replicates). Different letters indicate significant differences (*P* < 0.05, one-way ANOVA with two-sided Tukey's HSD test). *P* values are shown in the Source Data file. **c** ChIP-qPCR assay showed that the P1 region of the *Wx* promoter was significantly enriched for the transcription factor NF-YB1. **d** ChIP-qPCR assay showed that the P1 region of the *Wx* promoter was also significantly enriched by SLRL2. In (**c**, **d**), error bars represent SD (*n* = 3 independent experiments). Enriched values are normalized to the input. An unrelated *OsActin1* intron region was used as a negative control. IgG-immunoprecipitated DNA was used as a control. Statistical analysis was performed using a two-sided Student's *t* test. **e** EMSA analysis revealed the direct

interaction between NF-YB1 and the *Wx* promoter. **f** EMSA analysis showed that SLRL2 antagonized the binding activity of NF-YB1 to the *Wx* promoter. **g** Degradation analysis of NF-YB1 protein in the absence or presence of SLRL2 or its mutant form SLRL2△C3 protein. Protein samples were collected at the indicated time point for the following Western blot assay. The amount of NF-YB1-Myc protein was detected with an anti-Myc antibody, while the amount of SLRL2-Flag or SLRL2△C3-Flag protein was quantified with an anti-Flag antibody. The experiments in (**e**, **f**, **g**) were independently repeated three times with similar results. The source data are provided in a Source Data file. **h** Deletion of G-box, the NF-YB1 binding site, reduced the promoting effect of NF-YB1 on *Wx* expression but had no significant effect on the repressing effect of SLRL2 on *Wx* expression. **i** Determination of the regulation of *Wx* genes by NF-YB1 and SLRL2 in *nf-yb1* mutants. In (**h**, **i**), error bars represent SD (*n* = 5 biological replicates). Different lowercase letters above error bars indicate significant differences as determined by one-way ANOVA with Tukey's HSD test. *P* values are shown in the Source Data file.

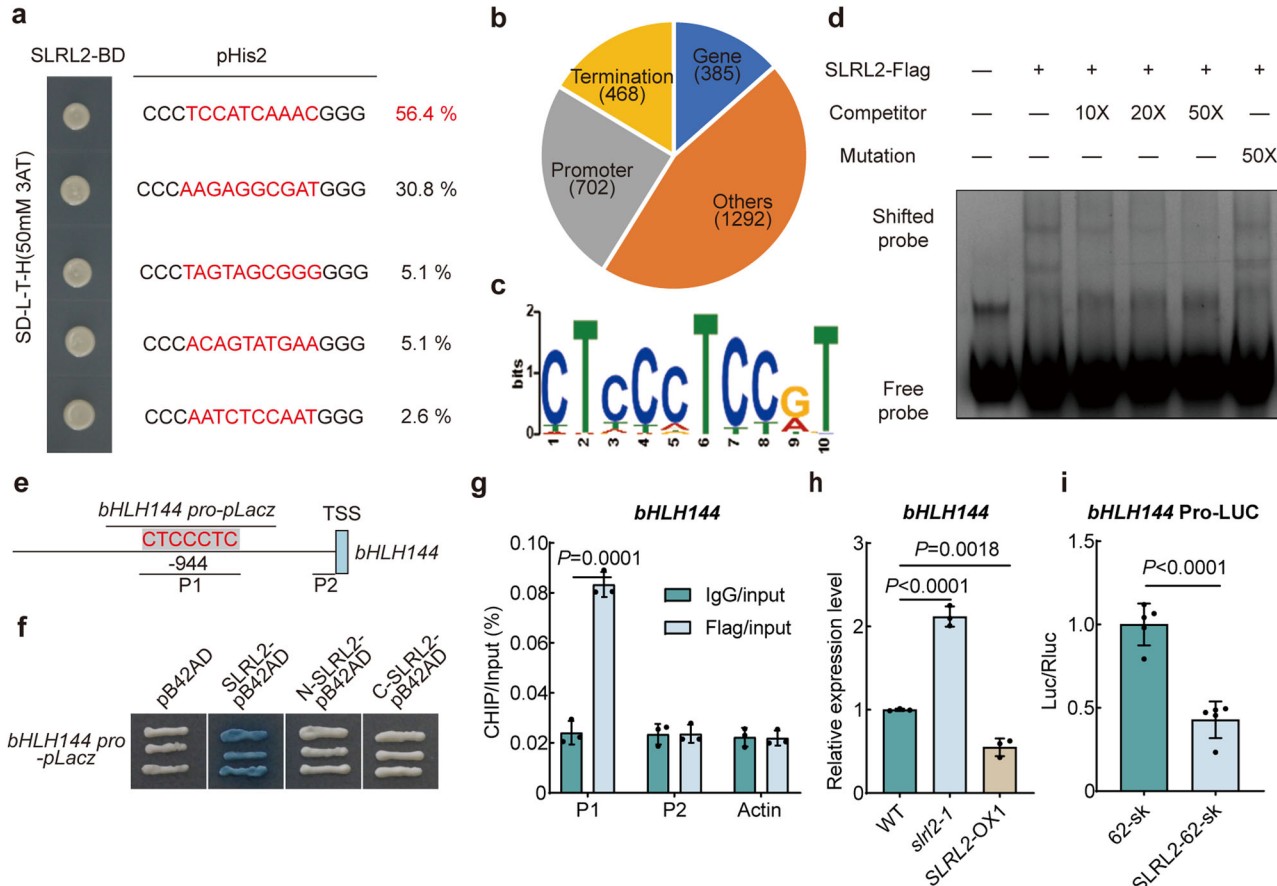

**Fig. 4 | SLRL2 directly represses the transcription of *bHLH144*. a** Yeast one-hybrid library screening to identify the potential SLRL2 binding site. Percentages represent the frequency of occurrence of the cis-acting element. **b** Distribution of SLRL2 binding sites in genic and intergenic regions revealed by DAP-seq analysis. **c** Conserved DNA binding site of SLRL2 identified by DAP sequencing. **d** EMSA analysis confirmed the direct binding of SLRL2 to the *bHLH144* promoter. The experiment was independently repeated three times with similar results. Source data are provided in a Source Data file. **e** Schematic representation of the *bHLH144* promoter and the potential SLRL2 binding site. **f** Yeast one-hybrid validation of the interaction between SLRL2 and the *bHLH144* promoter. **g** ChIP-qPCR assay showed that the P1 region of the *bHLH144* promoter was significantly enriched by SLRL2. Error bars represent SD (*n* = 3 experiments). Enriched values are normalized to the input. IgG-immunoprecipitated DNA was used as the control. **h** Expression analysis of *bHLH144* in *slrl2* mutant and *SLRL2* overexpressing transgenic rice. The RNA was extracted from the developing seeds at 15 DAF. The expression of *bHLH144* in the WT ZH11 was set to 1. Data are means ± SD (*n* = 3 biological replicates). **i** Investigation of the transcriptional regulation of *bHLH144* by SLRL2 using the dual-luciferase reporter assay. The value of the 62-SK and *bHLH144*pro-LUC co-transfection group was set to 1. Error bars represent SD (*n* = 5 biological replicates). Statistical analysis was performed by two-tailed Student's *t* test.

data give us some preliminary indication that SLRL2 may bind directly to the *Wx* promoter and regulate its expression.

### SLRL2 directly binds to *bHLH144* promoter and suppresses its expression

To identify the potential direct binding motifs of SLRL2, SLRL2 was used as a bait to screen a prey DNA library consisting of random short DNA sequences[24]. The result showed that SLRL2 can directly bind to several conserved DNA sequences. The abundance of each sequence was then calculated, and the two most abundant sequences accounted for 56.4% and 30.8% of the total sequenced clones, respectively (Fig. 4a). Analysis of these two sequences revealed that they both contained "CTCC/GAGG" (Fig. 4a). Next, the DNA affinity purification sequencing (DAP-seq) method was used to confirm that SLRL2 can bind directly to DNA[25]. Fortunately, a large number of SLRL2 target DNAs were identified, including more than 700 binding sites located in the gene promoter regions (Fig. 4b), fully demonstrating that SLRL2 indeed has direct DNA binding ability. By analyzing the enriched promoter sequences, the conserved "CTCC" sequence should be the cis-regulatory elements binding by SLRL2 (Fig. 4c), which was indeed the same as the yeast one-hybrid library screening result (Fig. 4a).

Interestingly, *bHLH144*, an important gene that responds to NF-YB1 stability, was included in the DAP-seq data. A "CTCC" sequence was indeed present in the *bHLH144* promoter (Fig. 4e). Previous reports have shown that bHLH144 can form a heterotrimer with the NF-YB1-YC12 protein complex, thereby contributing to the stabilization of NF-YB1[8]. Next, three approaches were used to verify the interaction between SLRL2 and *bHLH144* promoter. First, the EMSA result indicated that SLRL2 could directly bind to the wild-type *bHLH144* promoter instead of the one with a mutated CTCC binding motif (Fig. 4d). Second, the yeast one-hybrid analysis further demonstrated that only full-length SLRL2, and not its N-terminal or C-terminal split protein, can bind directly to the *bHLH144* promoter (Fig. 4f). Finally, ChIP-qPCR was employed to ascertain their in vivo association, and the P1 region of the *bHLH144* promoter was specifically highly enriched (Fig. 4g).

Next, the RT-qPCR result showed that *bHLH144* expression was increased in *slrl2* mutant, but decreased in *SLRL2* overexpression rice (Fig. 4h). The result of dual-luciferase reporter assay showed that SLRL2 significantly inhibited *bHLH144* transcription (Fig. 4i), which was consistent with the RT-qPCR result. Given the successful identification of a "CTCC" sequence in the P1 region of the *Wx* promoter, the EMSA experiment was used to confirm the potential direct

interaction between SLRL2 and the *Wx* promoter. The result demonstrated that SLRL2 could attach to the *Wx* probes and result in a distinct migratory band that could be diminished by rival probes (Supplementary Fig. 9). Therefore, the EMSA data provided solid evidence to demonstrate that SLRL2 could bind directly to the *Wx* promoter. Taken together, all these data indicated that SLRL2 is a canonical transcription factor with direct DNA binding ability and can repress the expression of both *bHLH144* and *Wx* genes, thereby modulating rice AC and grain quality.

### SLRL2 competes with NF-YC12 to interact with NF-YB1, thereby reducing the formation of the NF-YB1-YC12-bHLH144 heterotrimer and decreasing NF-YB1 stability

Since it is known that the interaction of NF-YC12 with NF-YB1 is required for the formation of the NF-YB1-YC12-bHLH144 heterotrimer complex, thereby stabilizing the NF-YB1 protein and consequently positively regulating *Wx* expression[8]. We wanted to understand how SLRL2 and NF-YC12 coordinate their interaction with the same NF-YB1 protein. Therefore, we first generated a series of NF-YB1 protein deletion constructs to determine which domain is responsible for its interaction with NF-YC12 and SLRL2, respectively (Fig. 5a). Yeast two-hybrid results indicated that the same NF-YB1 fragment (32-186aa) mediated its interaction with both SLRL2 and NF-YC12 (Fig. 5b, c). Then SLCA experiments confirmed that the involvement of SLRL2 protein attenuated the interaction intensity between NF-YB1 and NF-YC12 in vivo (Fig. 5d), suggesting that SLRL2 might compete with NF-YC12 to interact with NF-YB. Next, a detailed cell-free degradation analysis of NF-YB1 protein stability was performed to further investigate whether SLRL2 attenuates the formation of the NF-YB1-YC12-bHLH144 complex. The result indicated that SLRL2 or NF-YC12 alone or both proteins together can stabilize NF-YB1 protein in a similar manner (Fig. 5e, first four groups). While the formation of NF-YB1-YC12-bHLH144 complex can make NF-YB1 more stable (Fig. 5e, group six). Interestingly, further involvement of SLRL2 attenuates the effect of the NF-YB1-YC12-bHLH144 complex on NF-YB1 stability (Fig. 5e, group seven), suggesting that SLRL2 competes with NF-YC12 to interact with NF-YB1, thereby affecting the formation of the NF-YB1-YC12-bHLH144 complex. To further confirm this notion, the mutant SLRL2 protein (SLRL2ΔC3) with the NF-YB1 interaction domain deleted was added, and the NF-YB1 protein again became more stable (Fig. 5e, group eight). All these data show that SLRL2 competes with NF-YC12 for interaction with NF-YB1. Therefore, whether the formation of an NF-YB1-YC12-bHLH144 heterotrimer or an NF-YB1-SLRL2 heterodimer, with opposite roles in AC regulation, depends on the protein abundance of SLRL2 and NF-YC12. Finally, we examined the expression of NF-YC12 and SLRL2 in developing seeds with or without ABA treatment. The result showed that the expression of *NF-YC12* was only slightly increased, while that of *SLRL2* was remarkably promoted by ABA treatment (Fig. 5f). Taken together, less NF-YB1-YC12-bHLH144 complex is formed when the ABA level is increased, resulting in decreased rice AC.

### Direct inhibition of *SLRL2* expression by NF-YB1
Interestingly, we found that a G-box existed in the core region of *SLRL2* promoter (Fig. 6a). In addition, the RT-qPCR analysis showed that the *SLRL2* expression in the *nf-yb1* mutant was significantly increased (Fig. 6b). Then the result of dual luciferase reporter experiment showed that NF-YB1 inhibited *SLRL2* expression (Fig. 6c). Therefore, NF-YB1 should be a negative regulator of *SLRL2* transcription. To verify whether the regulation was realized by direct binding of NF-YB1 to the *SLRL2* promoter, three different experiments were performed. First, the yeast one-hybrid result proved that NF-YB1 can directly bind to the identified G-box in the *SLRL2* promoter (Fig. 6a, e). Next, the EMSA assay showed that NF-YB1 is strongly bound to the biotin-labeled *SLRL2* promoter fragment containing the G-box (Fig. 6d). Finally, the ChIP-qPCR data confirmed that the P1 region of the *SLRL2* promoter was

indeed significantly enriched for the NF-YB1 transcription factor (Fig. 6f). In conclusion, all the above results demonstrated that NF-YB1 could directly bind to the *SLRL2* promoter and inhibit its expression, thus further enriching the molecular regulatory network of rice quality, centered on the key NF-YB1-SLRL2-bHLH144 transcriptional regulatory cascade.

### *SLRL2* functions downstream of *NF-YB1* and *bHLH144* in the regulation of rice AC
To provide solid genetic evidence for this complicated regulatory module, we first generated *nf-yb1* and *bhlh144* rice mutants by using CRISPR/Cas9 technology. Two independent homozygous mutants with frameshift mutation of each gene were identified for subsequent analysis (Supplementary Fig. 10a, b). Phenotypic analyses showed that mutation of either *NF-YB1* or *bHLH144* did not alter major rice agronomic traits (Supplementary Fig. 10c–h). Assessing seed-related traits, the *bHLH144* knockout yielded no significant changes in the tested parameters (Supplementary Fig. 10i–l). Conversely, the *nf-yb1* mutants showed reduced grain length and width, hence the strongly reduced 1000-grain weight (Supplementary Fig. 10i–l).

Next, we focused on investigating the rice quality of *nf-yb1* and *bhlh144* mutants. Notably, both *NF-YB1* and *bHLH144* mutations resulted in a significant decrease in rice AC and an increase in rice GC (Supplementary Fig. 11a, b). Further expression analysis showed that *Wx* expression was decreased in the mutants, consistent with the change in AC (Supplementary Fig. 11c, d). DSC analysis showed that the onset, peak, and end gelatinization temperatures, and the gelatinization enthalpy were all significantly decreased in the *nf-yb1* and *bhlh144* mutants (Supplementary Fig. 11e). Finally, RVA analysis showed that *nf-yb1* and *bhlh144* mutants exhibited higher values of most viscosity parameters (Supplementary Fig. 11f). Further exploration involved generating double mutants, *slrl2 nf-yb1* and *slrl2 bhlh144*. Evaluation of AC and GC, two key traits for rice ECQ, displayed intriguing results. Although the AC of *nf-yb1* and *bhlh144* mutants was significantly lower than the WT control, the involvement of *slrl2* promoted their AC to the similar level as the *slrl2* single mutant, whose AC was significantly higher than the WT control (Fig. 6g). A similar result was obtained for the GC. That is, both the *slrl2 nf-yb1* and the *slrl2 bhlh144* double mutants had a GC similar to that of the *slrl2* mutant (Fig.6h). The expression of *Wx* was also consistent with the change in AC (Supplementary Fig. 11g, h). These data underscore SLRL2's role downstream of NF-YB1 and bHLH144 in regulating rice quality, particularly AC and GC.

### SLRL2, bHLH144 and NF-YB1 are all involved in the regulation of seed dormancy
The PHS of rice in the field was severe due to the continuous rainfall and high temperature during the rice harvest period. Interestingly, we found that the PHS of the *SLRL2*-overexpressing rice was significantly milder than that of the WT in the field (Fig. 7a). Therefore, we hypothesized that the newly discovered NF-YB1-SLRL2-bHLH144 module would similarly play a role in controlling rice seed dormancy. To better understand the fundamental parts of this molecular regulation module, we looked at the PHS phenotypes of the rice materials by using the PHS test system[24]. The results showed that *SLRL2* overexpression significantly reduced while *SLRL2* knockout slightly promoted rice PHS (Fig. 7b), suggesting that SLRL2 is a positive regulator of rice seed dormancy. Furthermore, the *SLRL2* mutation significantly attenuated the effect of ABA on rice PHS (Fig. 7c). As for *NF-YB1* and *bHLH144*, mutation of either gene also significantly reduced rice PHS (Fig. 7d–f), similar to that in the *SLRL2* overexpression rice. The *slrl2 nf-yb1* and *slrl2-bhlh144* double mutants both showed some degree of PHS resistance but were weaker than that of the *nf-yb1* or *bhlh144* single mutant (Fig. 7d–f). The above data showed that the key members of the identified NF-YB1-SLRL2-bHLH144 module are also involved in

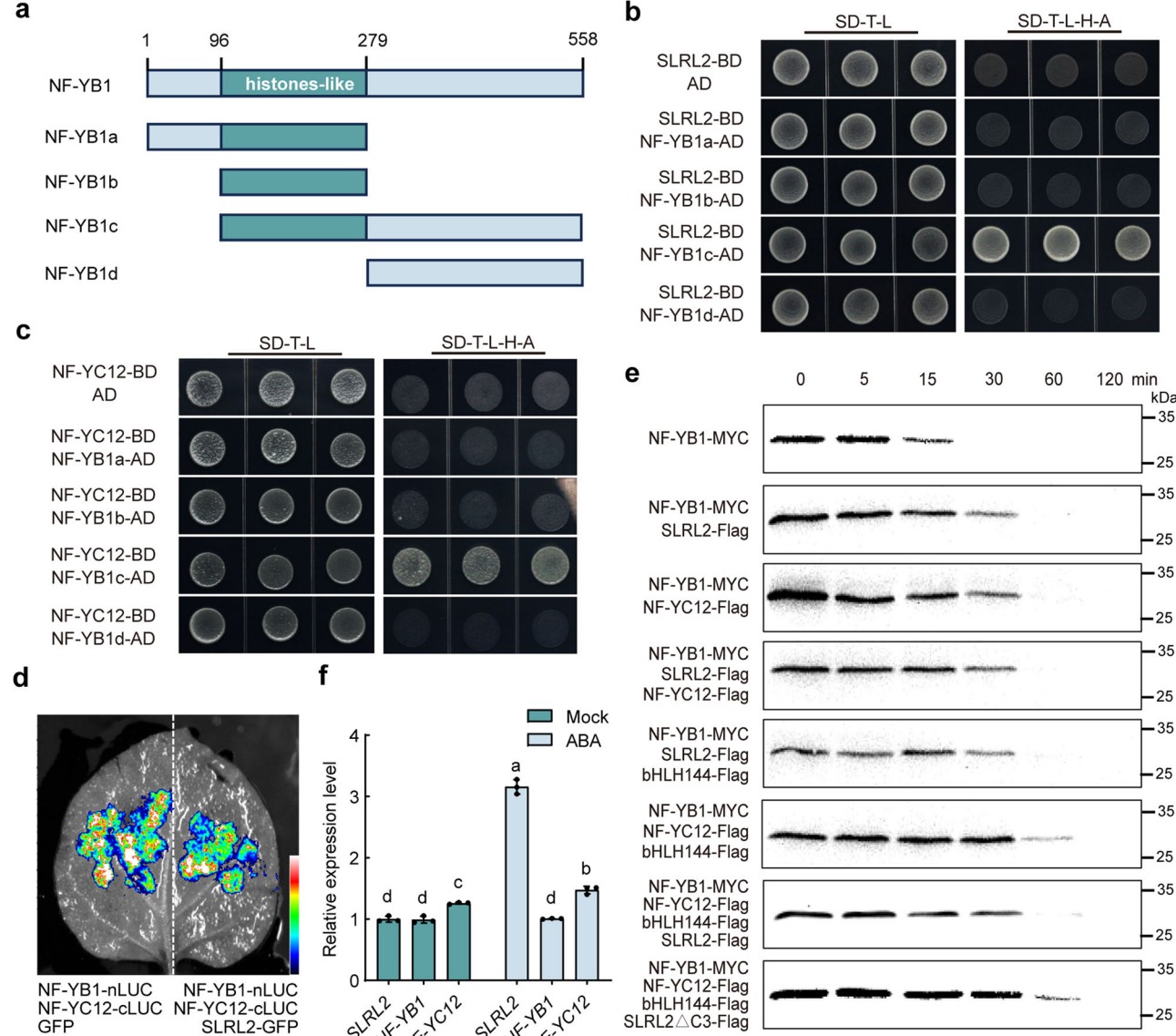

**Fig. 5 | SLRL2 competes with NF-YC12 to interact with NF-YB1. a** Diagrams for NF-YB1 protein truncations and functional motif. **b** Yeast two-hybrid assay for interactions between NF-YC12 and NF-YB1 truncations. **c** Yeast two-hybrid assay for interactions between SLRL2 and NF-YB1 truncations. **d** SLCA shows the inhibitory effect of SLRL2 on the interaction between NF-YC12 and NF-YB1. The fluorescence intensity in the image ranges from 1524 to 7024. **e** Cell-free degradation analysis of NF-YB1-MYC co-incubated with different protein sets, including SLRL2, NF-YC12, bHLH144, SLRL2ΔC3 and their different combinations. Protein samples were collected at the indicated time points for Western blot analysis. The amount of NF-YB1-

Myc protein was detected using an anti-Myc antibody. The experiments were independently repeated three times with similar results. Source data are provided in a Source Data file. **f** Expression analysis of *NF-YC12, SLRL2*, and *NF-YB1* in seeds (15 DAF) in response to ABA treatment. *OsActinO1* was used as an internal control to normalize the gene expression. The expression level of *NF-YC12* with mock treatment was set to 1. Data are means ± SD (*n* = 3 biological replicates). Different letters indicate significant differences (*P* < 0.05, one-way ANOVA with two-sided Tukey's HSD test). *P* values are shown in the Source Data file.

determining other seed-related growth and development events, including PHS in rice.

## SLRL2 affects rice seed dormancy by regulating *MFT2* expression

To elucidate the mechanism by which SLRL2 regulates rice dormancy, an in-depth analysis of the DAP-seq data was performed and the *MFT2* gene was identified as a potential target of SLRL2. MFT2 is a positive regulator of ABA signaling and suppresses rice seed germination[26]. Therefore, we first examined the expression of *MFT2* in the *SLRL2*-related rice materials. The result showed that the *MFT2* expression was significantly decreased in the *slrl2* mutant and slightly increased in the

*SLRL2* overexpression rice (Fig. 8a). The dual-luciferase reporter analysis also showed that SLRL2 could significantly activate *MFT2* expression (Fig. 8b). Further evidence from yeast one-hybrid assay, EMSA experiment, and ChIP-qPCR analysis all showed that SLRL2 can directly bind to the promoter of *MFT2* in vitro and in vivo (Fig. 8c–f). Taken together, these results suggest that SLRL2 regulates seed germination and dormancy, at least in part, by directly activating the expression of *MFT2*. Since no PHS data were reported in the previous studies, we generated *mft2* mutants using the CRISPR/Cas9 method to further confirm this conclusion (Supplementary Fig. 12). As expected, PHS analysis showed that the *MFT2* mutation significantly increased the PHS of rice (Fig. 8g, h).

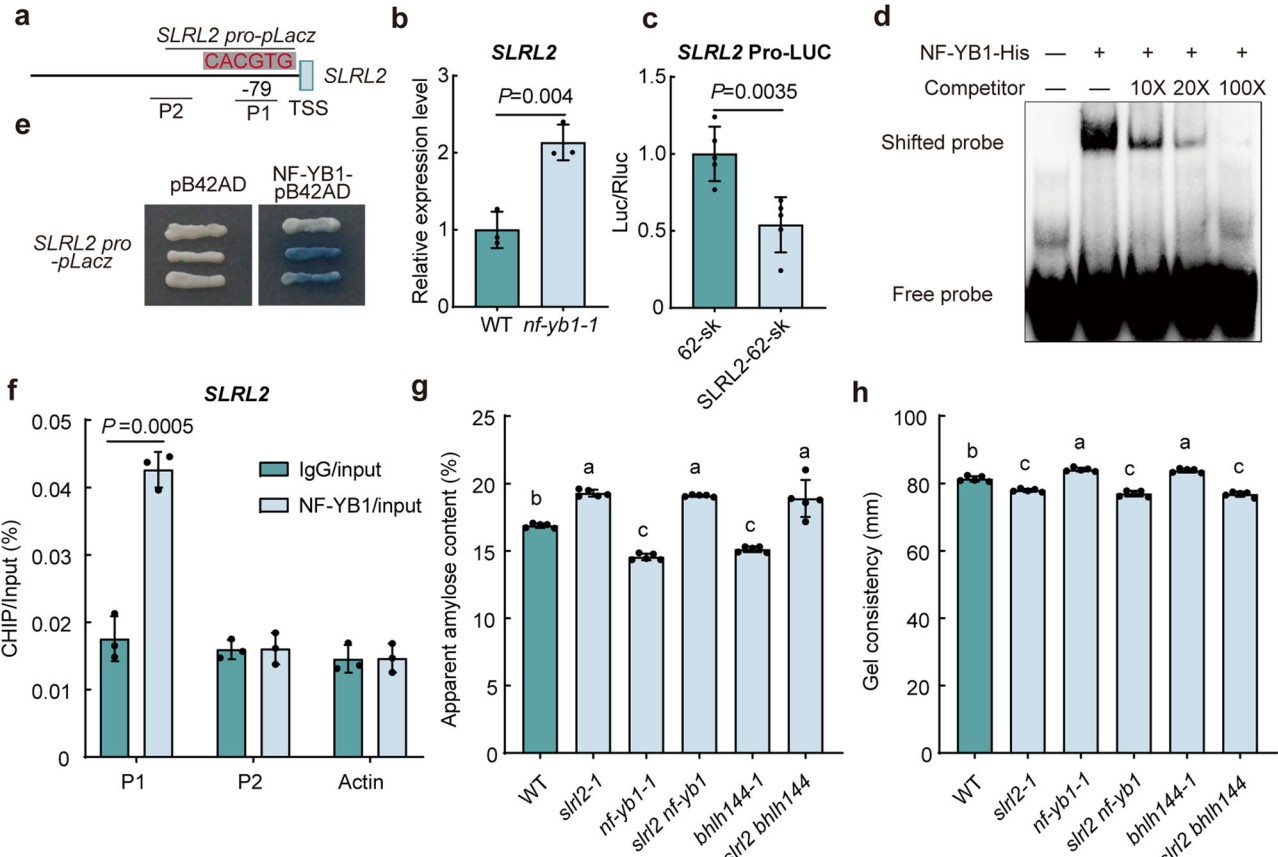

**Fig. 6 | NF-YB1 directly represses the transcription of *SLRL2*. a** Structural diagram of the NF-YB1 binding G-box on the *SLRL2* promoter. **b** Expression analysis of *SLRL2* in *nf-yb1* mutant. *OsActin1* was used as an internal control to normalize gene expression. The RNA was extracted from the developing seeds at 15 DAF. The expression level of *SLRL2* in WT was set to 1. Data are means ± SD (*n* = 3 biological replicates). **c** Dual luciferase reporter analysis showing the suppressive effect of NF-YB1 on *SLRL2* expression. The dual fluorescence ratio of *SLRL2*pro-LUC combined with 62-SK was set to 1. Data are means ± SD (*n* = 5 biological replicates). **d** EMSA experiment confirming the direct interaction between NF-YB1 and the *SLRL2* promoter. The hot probe was a biotin-labeled *SLRL2* promoter fragment with a G-box motif, while the competitor probe was unlabeled. **e** Yeast one-hybrid experiments showing the direct interaction between NF-YB1 and the *SLRL2* promoter. **f** The ChIP-qPCR assay showing the enrichment of the *SLRL2* promoter by NF-YB1 in vivo. Data are means ± SD (*n* = 3 independent experiments). In (**b**, **c**, **f**), statistical analysis was performed using two-tailed Student's *t* test. Analysis of AC (**g**) and GC (**h**) of the single and double rice mutants concerning *SLRL2*, *NF-YB1*, and *bHLH144* genes. Data are means ± SD (*n* = 5 biological replicates). Different lowercase letters above error bars indicate significant differences (*P* < 0.05, one-way ANOVA with two-sided Tukey's HSD test). *P* values are shown in the Source Data file.

## Discussion

Rice seed development, in particular grain shape and rice quality, is regulated by dynamic phytohormone levels. As a well-known stress hormone, ABA levels in seeds are not only correlated with the seed dormancy trait, but are also closely correlated with other seed-related traits[27–32]. During abiotic stress, ABA levels in plants increase rapidly, accelerating plant senescence and stimulating nutrient recycling from source tissues to reproductive tissues to ensure seed reproduction[33–35]. Fast delivery of enough nutrients to seeds will hasten the production of seed storage components[36]. Therefore, appropriate drought or ABA treatment can promote the grain-filling speed of both rice and wheat[37,38].

There are, however, very few reports on how ABA impacts rice ECQ, which is the most significant consumer feature of rice. This study showed that ABA treatment suppressed *Wx* gene expression and consequently reduced rice AC, mediated by an uncharacterized transcriptional regulator SLRL2 (Figs. 1a–e and 3b, d). Like ABA, SLRL2 is a negative regulator of rice AC. In particular, *SLRL2* overexpression decreased, while its deletion enhanced *Wx* expression and rice AC (Fig. 1j and Supplementary Fig. 6b), which is different from most other reported transcription factors with positive regulatory roles on the *Wx* gene[7,9,39]. Furthermore, rice AC is also sensitive to environmental

stress, such as high temperature and salt stress, both of which will result in reduced AC[40,41]. Considering the pleiotropic effects of ABA, especially on rice AC and abiotic stress response, the revealed molecular mechanism of ABA in regulating *Wx* expression will be beneficial for future breeding of elite rice varieties with comprehensive traits improved by fine modulation of the ABA pathway. Editing non-coding regulatory DNA of key target genes is becoming an effective and promising strategy to achieve this goal[42].

NF-YB1 is an important seed-specific nuclear factor with multiple roles in different events of rice seed development, including grain size, endosperm chalkiness, cell cycle, sugar loading, starch biosynthesis, and grain filling[8,10,43–46]. Extensive experiments demonstrated the direct interaction between SLRL2 and NF-YB1 (Fig. 2). Moreover, SLRL2 can inhibit the degradation of NF-YB1, while the deletion of SLRL2's PFYRE domain eliminated this effect (Fig. 3g). A similar protective role has also been reported for the DELLA protein SLR1, which binds to MONOCULM 1 (MOC1) to prevent its degradation, thereby regulating tiller number and plant height in rice[47]. Surprisingly, SLRL2 also directly inhibits the transcription of *bHLH144*, a key regulator that stabilizes the NF-YB1 protein against degradation. Interestingly, SLRL2 also competes with NF-YC12 to interact with NF-YB1, thereby affecting the formation of the NF-YB1-YC12-bHLH144 complex, and

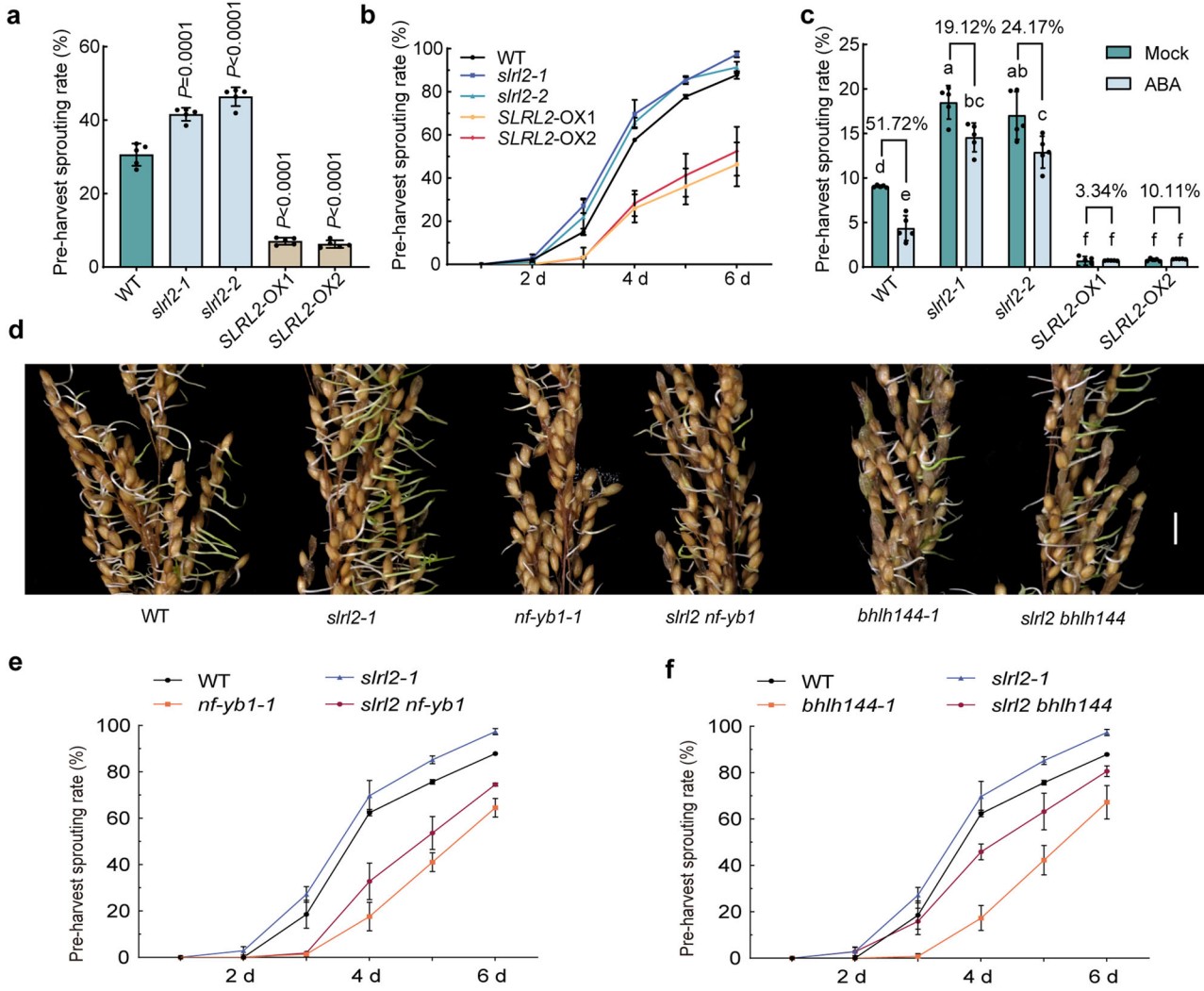

**Fig. 7 | *SLRL2*, *NF-YB1*, and *bHLH144* are all involved in the regulation of pre-harvest sprouting (PHS) in rice.** Field PHS data (**a**) and PHS analysis (**b**) of *slrl2* mutants and transgenic rice overexpressing *SLRL2*. Data are means ± SD (*n* = 5 biological replicates, two-sided Student's *t* test). **c** Field PHS data of *SLRL2*-related materials in response to ABA treatment. Data are means ± SD (*n* = 5 biological replicates). Different letters denote significant differences (*P* < 0.05, one-way ANOVA with two-sided Tukey's HSD test). *P* values are shown in the Source Data file. **d** PHS phenotypes of *slrl2*, *nf-yb1*, and *bhlh144* single and double mutants. The scale bar is 1 cm. **e** PHS analysis of *nf-yb1*, *slrl2*, and *slrl2 nf-yb1* double mutants. **f** PHS analysis of *bhlh144*, *slrl2*, and *slrl2 bhlh144* double mutants. Data are means ± SD (*n* = 5 biological replicates).

consequently decreasing NF-YB1 abundance and rice AC (Fig. 5). Finally, NF-YB1 binds directly to the promoter of *SLRL2* and represses its transcription. Thus, SLRL2, NF-YB1, and bHLH144 form a complex regulatory module characterized by reciprocal interactions at multiple levels, including transcriptional control, modulation of transactivation activity, and regulation of protein stability (Fig. 9).

Furthermore, SLRL2 is not only directly involved in the regulation of rice AC and ECQ, but also functions in the modulation of rice dormancy and PHS. *SLRL2* overexpression increased PHS resistance, whereas *SLRL2* mutation only slightly decreased rice resistance to PHS (Fig. 7a, b). *MFT2*, a key positive regulator of ABA signaling, was the direct target of SLRL2. SLRL2 regulates rice PHS, at least in part, by modulating *MFT2* expression. Therefore, this result prompted us to further investigate whether modulation of *NF-YB1* and *bHLH144*, two other key components of this module, would also affect rice PHS. Interestingly, knockout of either *NF-YB1* or *bHLH144* also significantly reduced rice PHS (Fig. 7d−f), a phenomenon not previously reported but consistent with that of the *SLRL2*-overexpressing rice. Thus, the results of the PHS assays suggested that SLRL2, bHLH144 and NF-YB1 all participate in PHS regulation.

Interestingly, SLRL2 was considered to be a member of the DELLA family proteins based on phylogenetic analysis[48]. Furthermore, SLRL2 also shares similar subcellular localization, transactivation activities and plant height suppressive effects with SLR1 (Supplementary Figs. 4d−h and 5e, f). Nevertheless, the two proteins show different expression patterns, DNA binding activities, and responses to hormonal stimuli. For example, *SLRL2* transcription was induced by ABA (Fig. 1f−h), but both its transcription and protein stability were not affected by GA (Supplementary Fig. 4a−c), suggesting that SLRL2 is a potential component of the ABA pathway instead of GA. Most importantly, we demonstrated that SLRL2 has a DNA-binding activity that is quite different from that of the DELLA proteins. To date, there is no credible evidence that the DELLA protein can bind directly to the promoters of target genes. Instead, the DELLA protein always indirectly modulates the expression of downstream target genes by interacting with other transcription factors[22,23,49,50]. In this study, the conserved CTCC sequence was identified as the ideal binding site for SLRL2. Then, a series of analyses further confirmed that SLRL2 can directly bind to the promoters of *Wx* and *bHLH144* and thereby regulate their expression.

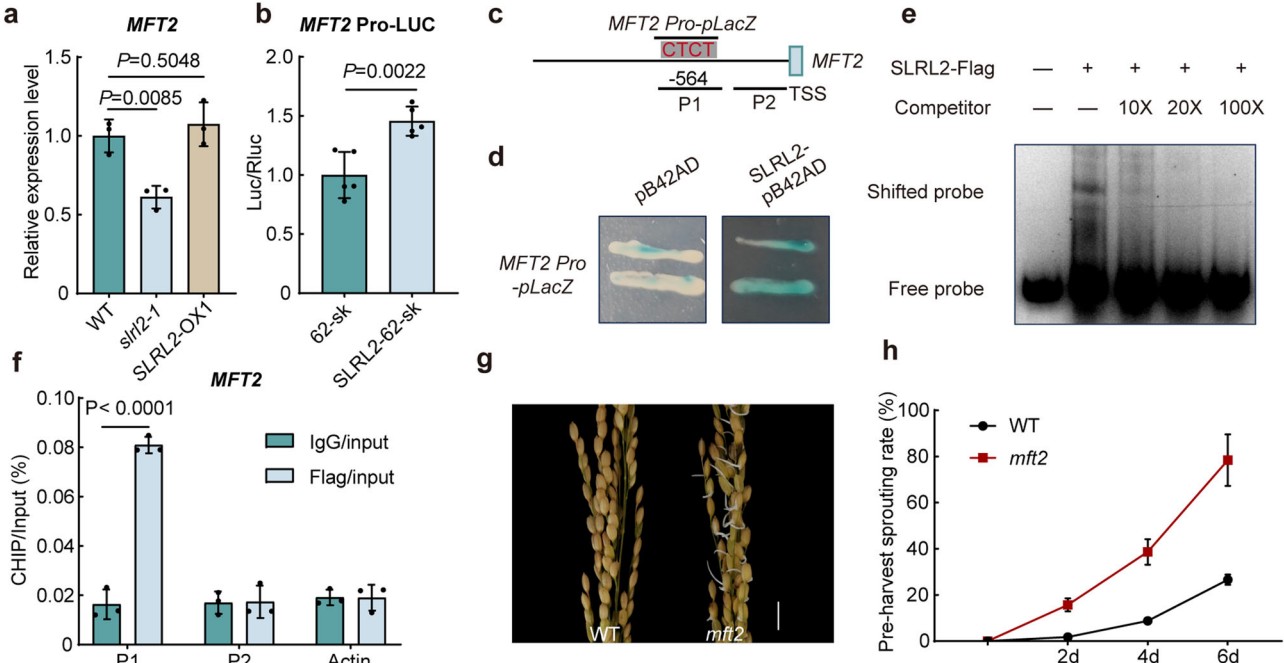

**Fig. 8 | SLRL2 directly activates the transcription of *MFT2* to inhibit rice PHS.** **a** Expression analysis of *MFT2* in *slrl2* mutant and *SLRL2* overexpression transgenic rice. *OsActin1* was used as an internal control to normalize gene expression. The expression level of *MFT2* in WT was set to 1. Data are means ± SD ($n = 3$ biological replicates). **b** Dual-luciferase reporter assay showed that SLRL2 activates the expression of *MFT2*. The value of the 62-SK and *MFT2*pro-LUC co-transfection group was set to 1. Data are means ± SD ($n = 5$ biological replicates). A two-sided Student's *t* test was performed for statistical analysis. **c** Structural diagram of the SLRL2-binding cis-acting element on the *MFT2* promoter. **d** Yeast one-hybrid experiments showing the direct interaction between SLRL2 and the *MFT2* promoter. **e** EMSA experiment confirming the direct interaction between SLRL2 and the *MFT2* promoter. The hot probe is a biotin-labeled *MFT2* promoter fragment with a CTCC motif, while the competition probe is unlabeled. The experiments were independently repeated three times with similar results. Source data are provided in a Source Data file. **f** The ChIP-qPCR assay shows the enrichment of the *MFT2* promoter by SLRL2 in vivo. Data are means ± SD ($n = 3$ independent experiments). Statistical analysis was performed using two-tailed Student's *t* test. Panicle phenotypes (**g**) and quantification curve (**h**) of PHS analysis of the *mft2* mutant and its WT. The scale bar is 1 cm. Error bars represent SD ($n = 5$ biological replicates).

DELLA proteins, as central negative regulators of the GA signaling pathway, were the major contributors to the Green Revolution in the 1960s[51]. A classical GA–GID1–DELLA core complex for the GA signaling pathway is successfully established[22,52,53]. As a result, it is now widely accepted that the regulation of the DELLA protein is tightly controlled by GA signaling. However, the actual evolutionary path of the GA signaling pathway remains elusive. Our present study further strengthens the hypothesis centered on an upside-down perspective of hormone signaling, i.e., that DELLA proteins predate the GA perception by GID1[54]. This view is supported by two key lines of evidence. The first one is that the existence of other signals, such as light and temperature, can regulate the accumulation of DELLA proteins separate from GA, which process is mediated by the ubiquitin E3 ligase COP1 and the proteasomal degradation pathway[55]. Furthermore, phylogenetic analysis and research into the roles of particular genes also provide another important source of data. This data emphasizes the existence of DELLA orthologous proteins without the typical DELLA domain, demonstrating that these proteins function outside of the conventional GA-GID1-DELLA regulatory module framework. Nevertheless, the amino acid sequence similarity of the GRAS domains between these uncanonical DELLA-like proteins and DELLA proteins was sufficient to establish their closest phylogenetic relationships.

Here, we identified a seed-specific DELLA family member, SLRL2, with direct DNA-binding activity. We then successfully established a complex and distinct regulatory module, NF-YB1-SLRL2-bHLH144, that is central to mediating ABA regulation of rice quality. In addition, we demonstrated that SLRL2, bHLH144, and NF-YB1 also function in the regulation of rice dormancy and PHS. Our findings shed new light on how to improve multiple seed traits by modulating the ABA signaling pathway, thereby strengthening rice yield and quality assurance.

## Methods

### Plant materials and growth conditions

All rice materials used in this study were under the genetic background of the japonica cultivar Zhonghua11 (ZH11) and were cultivated in paddy fields or greenhouses at Yangzhou University (Yangzhou, China). All rice plants were grown under the same climatic conditions and crop management practices. At the seed maturity period, the major agronomic traits of rice were measured, including plant height, number of tillers, flag leaf length, and width. During the seed maturity stage, all major seed-related traits, such as seed setting rate, seed size, and 1000 grain weight, were analyzed carefully.

### Generation of transgenic rice plants and the double mutants

To generate rice mutants employing the CRISPR/Cas9 gene editing method, specific target sites for the genes *SLRL2*, *NF-YB1*, *bHLH144*, *NCED2*, *DG1* and *MFT2* were carefully designed and cloned into the *pC1300-Cas9* vector, respectively[56]. Upon transformation into the recipient rice cultivar ZH11, the editing target loci were sequenced to detect the mutation types. The construction of the *SLRL2* overexpression vector entailed cloning the *SLRL2* coding sequence (CDS) into the binary vector *pCAMBIA1300* driven by the *OsActin01* promoter and fused with a *Flag* epitope. The mutant lines *slrl2-1*, *nf-yb-1* and *bhlh144-1* were used to generate the double mutants *slrl2 nf-yb1* and *slrl2 bhlh144*. All primers used in this study are listed in Supplementary Data 2.

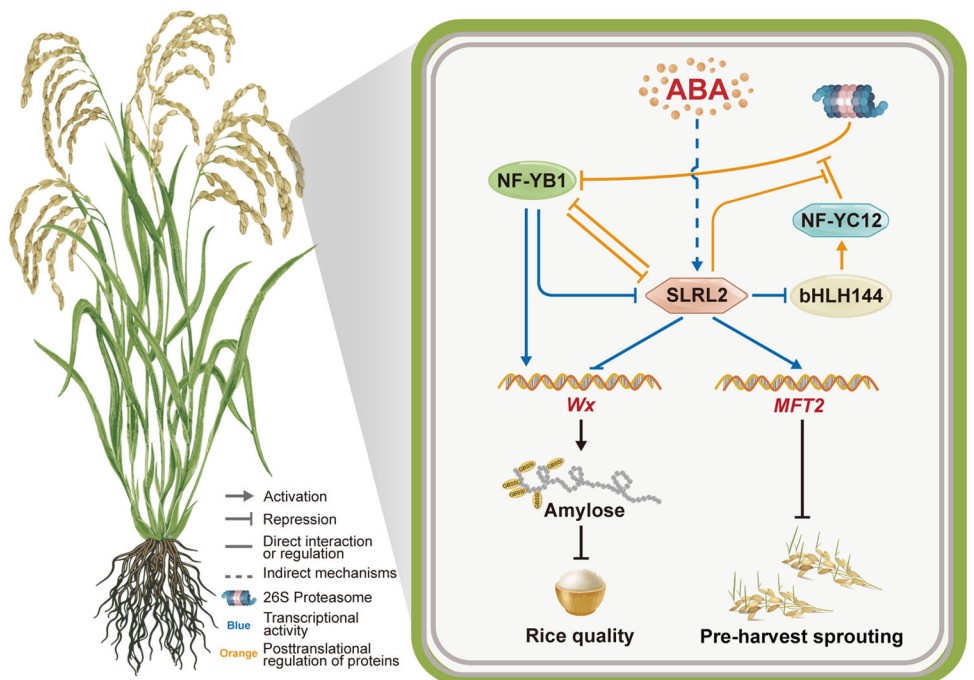

**Fig. 9 | A model for the NF-YB1-SLRL2-bHLH144 regulatory cascade in rice AC and PHS.** At stages of rice seed development or under environmental stress, ABA levels are elevated in the seed, which induces the expression of *SLRL2*. SLRL2 is a DELLA family member with both transcriptional activation activity and direct DNA binding activity. Specifically, SLRL2 is a negative regulator of rice AC by repressing the expression of the *Wx* gene, which is the primary key determinant of rice AC. SLRL2 fine-tunes *Wx* gene expression through a variety of approaches. First, SLRL2 binds directly to the *Wx* promoter and represses its expression. Second, SLRL2 physically interacts with and represses the transcriptional activity of NF-YB1, a key positive regulator of *Wx* gene expression. Third, SLRL2 binds directly to the *bHLH144* promoter and represses its expression. It is known that bHLH144 can increase the stability of NF-YB1 by forming a heterotrimer complex with NF-YB1 and NF-YC12. Fourth, SLRL2 interacts with NF-YB1 to form a heterodimer, which is also

helpful in increasing NF-YB1 stability. Finally, SLRL2 competes with NF-YC12 to interact with NF-YB1, thereby affecting the formation of the NF-YB1-YC12-bHLH144 heterotrimer and decreasing NF-YB1 stability. Thus, a SLRL2-NF-YB1-bHLH144 transcriptional regulatory module is established for the fine regulation of rice AC. In addition, this molecular work module is also suitable for the regulation of rice PHS. SLRL2 binds directly to the *MFT2* promoter and activates its expression. MFT2 is a positive regulator of ABA signaling and promotes seed dormancy. Therefore, the SLRL2 centered-regulatory module may also mediate ABA-regulated seed dormancy. Arrows and bars indicate activation and repression effects, respectively. Orange color indicates post-translational regulation. Blue color indicates transcriptional regulation. Solid and dashed lines indicate direct and indirect (or unknown mechanism) effects, respectively.

### RNA extraction and RT-qPCR
Total RNA was extracted from various rice tissues using the FastPure Universal Plant Total RNA Isolation Kit (Vazyme Biotech, Jiangsu, China). HiScript II Q Select RT SuperMix (Vazyme Biotech, Jiangsu, China) was used for cDNA synthesis. The expression of each gene was quantified by RT-qPCR using the CFX Connect Real-Time PCR Detection System (Bio-Rad, California, USA) and AceQ qPCR SYBR Green Master Mix (Vazyme Biotech, Jiangsu, China). The housekeeping gene *OsActin01* served as an internal control, and each experiment was conducted with a minimum of three biological replicates.

### In situ hybridization analysis
Digoxigenin-labeled in situ hybridization was performed as previously described with minor modifications[57]. The hybridization probe was amplified and then cloned into the *pEASY-T3* vector (Transgene Biotech, Beijing, China). Labeling of the probe was accomplished by using the Digoxin RNA Labeling Kit (Roche, Basel, Switzerland) following the manufacturer's standard protocol. Hybridization was performed overnight at 42 °C with the solution as previously described[57]. An alkaline phosphatase assay-labeled probe coupled to an anti-DIG antibody Fab fragment and 4-nitroblue tetrazolium chloride/5-bromo-4-chloro-3-indolyl phosphate (Roche, Basel, Switzerland) was used for the experiment. Tissue sample sections were subjected to ethanol series for dehydration, stained, and then cleared in xylene.

### Yeast one-hybrid assay
In this study, the Clontech™ one-hybrid system (Clontech, Dalian, China) was used. The CDS of the potential trans-activator was fused to the GAL4 AD structural domain in the *pB42AD* vector (Clontech, Dalian, China), and the promoter sequences of *Wx*, *SLRL2*, and *bHLH144* genes were cloned into *pLacZi* vector (Clontech, Dalian, China), respectively. Yeast strain EGY48 was transformed and cultivated on the corresponding SD dropout medium for selection (Clontech, Dalian, China). Protein-DNA interactions were confirmed by the detection of blue colonies on the selection medium.

### Yeast two-hybrid assay
For yeast two-hybrid assays, the CDS and truncated fragments of target genes, such as *SLRL2*, *NF-YB1*, and *bHLH144*, were amplified and cloned into *pGADT7* (AD) or *pGBKT7* (BD) vectors, respectively. The AD and BD fusion constructs were co-transformed into yeast strain AH109 using the EX-Yeast Transformation Kit (Zoman, Beijing, China). The transformants were grown on the SD/-Leu-Trp solid medium at 30 °C for 3 days. Then the transformants were selected on the SD/-Leu-Trp-His solid medium with appropriate concentrations of 3-amino-1, 2, and 4-triazole (3-AT) for three to five days to study the interactions[47].

### Split luciferase complementation assays (SLCAs)
The *JW771* and *JW772* vectors were used for firefly luciferase (LUC)-based SLCAs[10]. The ORFs of *SLRL2* and *NF-YB1* were cloned into these vectors, respectively. *Agrobacterium tumefaciens* (GV3101) cells

carrying the constructed plasmids were resuspended (OD600 = 1.5) in infiltration buffer (10 mM MgCl$_2$ and 200 μM acetosyringone). Equal volumes of LUCn and LUCc suspensions were mixed and infiltrated into *Nicotiana benthamiana* leaves. Approximately 36 h post-infiltration, the leaves were sprayed with 0.32 mg/ml D-fluorescein potassium salt in 0.1% (v/v) Triton X-100, and the fluorescence signal was detected using a CCD camera (Uvitec Alliance Q9).

### BiFC assay
The BiFC assay was conducted with some modifications based on the detailed procedure previously published[10]. The CDS of *SLRL2* and *NF-YB1* were cloned into *pXY104-cYFP* and *pXY103-nYFP* vectors, respectively. The plasmids were then co-transformed into rice protoplast cells, and the cells were incubated overnight at 25 °C in the dark. Finally, the fluorescence signals were observed under a confocal laser scanning microscope (LSM 710; Carl Zeiss AG, Germany). Primer information is provided in Supplementary Data 2.

### Antibodies used in this study
The following antibodies were used for the western blotting experiments, primary antibodies: anti-Flag monoclonal antibody (Sigma-Aldrich, Cat.#F3165, 1:8000 dilution), anti-Hsp82 monoclonal antibody (Beijin Protein Innovation (BPI), Cat.#AbM51099-31-PU, 1:5000 dilution), anti-Myc antibody (TransGen Biotech, Cat.#HT101, 1:3000 dilution), anti-GFP Mouse Monoclonal Antibody (TransGen Biotech, Cat.#HT801, 1:3000 dilution), anti-OsGBSSI Rabbit Monoclonal Antibody (ABclonal, Cat.#A19151, 1:3000 dilution). Secondary antibodies: Goat Anti-Mouse IgG-HPR secondary antibody (CWBIO, Cat.#CW0102S, 1:8000 dilution), Goat Anti-Rabbit IgG/HRP secondary antibody (Solarbio, Cat.#SE134, 1:8000 dilution).

### Pull-down assay
The CDSs of *SLRL2* and *NF-YB1* were cloned into the *pKL293-MFH* vector, respectively, yielding the plasmids *SLRL2-pKL293-Flag* and *NF-YB1-pKL293-MYC*. The TnT® SP6 High-Yield Wheat Germ Protein Expression System (Promega) was used to express the target proteins. The expressed tag proteins MYC and Flag were used as negative controls. The pull-down experiment was performed following our previously published protocol[58], with primer details listed in Supplementary Data 2.

### Co-immunoprecipitation (Co-IP) analysis
DNA constructs encoding SLRL2-Flag and NF-YB1-Myc recombinant proteins were generated. Then, three sets of plasmid combinations, including SLRL2-Flag and Myc, Flag and NF-YB1-Myc, SLRL2-Flag and NF-YB1-Myc, were co-transformed into rice protoplasts, respectively. After overnight incubation, total protein was extracted with anti-Flag antibody-conjugated agarose beads. Proteins pulled down by SLRL2-Flag were separated by SDS-PAGE and detected with anti-Myc antibody. SLRL2-Flag was detected with the anti-Flag antibody.

### EMSA analysis
The target gene of *SLRL2* or *NF-YB1* was cloned into the *pKL293-MFH* vector. 5 μg of the constructed plasmid was added to 15 μL of Promega Wheat Germ Expression MIX, and incubated at 28 °C for 2 h. Protein profiles were detected using Western blot. Complementary oligonucleotide chains were synthesized in the Dynactin section, and the oligonucleotide chains were labeled using the Beyotime Biotin Labeling Kit and then annealed to form a double strand. The EMSA experiment was performed according to the method provided in the EMSA chemiluminescence kit (Beyotime, China).

### Dual-luciferase reporter assay
The promoters of *Wx* and *bHLH144* were cloned into the *pGreenII 0800-LUC* vector to create reporter constructs. The CDS of either

*SLRL2* or *NF-YB1* was cloned into two vectors, the *pGreenII BD* and *pGreenII 62-sk*, respectively. Thus, *Wx*pro-LUC, *bHLH144*pro-LUC, *NF-YB1*-pGreenII BD, *SLRL2*-pGreenII BD, *NF-YB1*-pGreenII 62-sk and *SLRL2*-pGreenII 62-sk plasmids were successfully constructed. Plasmid combinations were co-transformed into rice protoplast based on the experimental design. The transformed cells were incubated for 12 h at 28 °C in the dark. *SLR1*-pGreenII BD was used as a positive control for transcriptional activity analysis. Transcriptional activity was measured by using a dual luciferase assay kit (Vazyme Biotech, Jiangsu, China).

### Chromatin immunoprecipitation (ChIP)-qPCR
The ChIP assay was performed as previously described[27]. Briefly, 7 day-after-flowering (DAF) developing seeds of the *SLRL2-3Flag* overexpression transgenic rice were used for the analysis. Anti-Flag monoclonal antibody (Sigma, USA) conjugated to immunomagnetic beads was used to immunoprecipitate the protein-DNA complex. The precipitated DNA fragments were analyzed by RT-qPCR to evaluate the enrichment. Primer sequences for this experiment are provided in Supplementary Data 2.

### Analysis of rice grain quality
Mature rice seeds were harvested, dried, chaffed, and polished using a grain polisher (Kett, Tokyo, Japan). The edible properties of the seeds were measured using a grain analyzer (Perten IM9500, Sweden). After measuring the palatability, the polished particles were further ground to powder using a FOSS 1093 Cyclotec Sample Mill (Tecator, Höganäs, Sweden) for the analysis of the physicochemical properties of the grain. Apparent amylose content (AAC), gel consistency (GC), pasting properties, and gelatinization properties were measured according to previously published methods 1[4]. The pasting properties of starch were determined using a differential scanning calorimeter (DSC 200 F3, Netzsch Instruments NA LLC; Burlington, MA). The pasting properties of rice were evaluated using a rapid viscosity analyzer (RVA) (Techmaster, Newport Scientific, Warriewood, Australia)[59].

### Rice PHS analysis
To assess the PHS traits, superior mature panicles from the *SLRL2, NF-YB1*, and *bHLH144*-related rice mutants, as well as from the WT ZH11, were used. These evaluations were conducted under laboratory conditions that mimic the natural high temperature and high humidity conditions that readily stimulate rice PHS in the field. The freshly collected panicles were treated with saturated water and incubated at 30 °C, and the number of germinated seeds was recorded at two-days intervals[27].

### Cell-free degradation assay
The analysis was performed according to the procedure outlined by Bello et al. (2014) with minor modifications. Initially, the degradation buffer (25 mM Tris-HCl, pH 7.5,10 mM NaCl, 10 mM MgCl2, 4 mM PMSF, 5 mM DTT, and 10 mM ATP) was used to extract the total protein from 10-day-old rice seedlings. The extracted protein was quantified by using a Quabit system (Invitrogen, Carlsbad, CA). Then purified recombinant proteins, 5 μg for each protein sample, were incubated with the extracted total proteins (200 μg) in the degradation buffer (20 μL) at 28 °C. Samples were collected at specified time intervals, and the degradation reactions were stopped immediately. The amount of the target protein was determined by Western blotting using anti-His, anti-Flag and anti-MYC antibodies.

### Statistics and reproducibility
Statistical analysis and number of biologically independent sample (n) are indicated in the figure legends. Comparisons were made by two-tailed Student's *t*-test using Microsoft Excel 2021 software. One- or two-way ANOVA with two-sided Tukey's HSD test using GraphPad Prism 8 software.

## Accession numbers

Sequence data from this article can be found in the GenBank Database and rice genome annotation project (RAPDB)under the following accession numbers: *SLRL2* (LOC_Os05g49930); *SLR1* (LOC_Os03g49990); *SLRL1* (LOC_Os01g45860); *NF-YB1* (LOC_Os02g49410); *NF-YC12* (LOC_Os10g11580); *bHLH144* (LOC_Os04g35010); *NAC20* (LOC_Os01g01470); *NAC26* (LOC_Os01g29840); *bZIP58* (LOC_Os07g08420); *MADS14* (LOC_Os03g54160); *Wx* (LOC_Os06g04200); *NCED2* (LOC_Os12g24800); *DG1* (LOC_Os03g12790); *MFT2* (LOC_Os01g02120); *NF-YC9* (LOC_Os01g01290).

## Reporting summary

Further information on research design is available in the Nature Portfolio Reporting Summary linked to this article.

## Data availability

All the relevant data supporting the results of this work are available both in this article and in the Supplementary Information files. All constructs and transgenic plants are available upon request. Gene sequences are available from RAPDB using the accession numbers provided in this article. Source data are provided with this article. Source data are provided with this paper.

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

## Acknowledgements

We thank Jian Zhang (China National Rice Research Institute) for providing the NF-YB1 antibody and Yucheng Wang (Northeast Forestry University) for providing the yeast one-hybrid library. This work was supported by the STI 2030-Major Project (2023ZD0406902, Recipient: Q.-F.L.); the National Natural Science Foundation of China (32270575, Recipient: Q.-F.L.; 32201740, Recipient: L.-C.H.); the Project of Zhongshan Biological Breeding Laboratory (BM2022008-02, Recipient: Q.-Q.L.); the Programs from Government of Jiangsu Province (JBGS[2021]001, Recipient: Q.-Q.L.; BK20200045, Recipient: Q.-F.L.; BK20220567, Recipient: L.-C.H.; 22KJB210005, Recipient: L.-C.H.; PLR202101, Recipient: C.C.; PAPD); and Post-graduate Research & Practice Innovation Program of Jiangsu Province (KYCX21_3237, Recipient: J.-D.W.; KYCX21_3249, Recipient: J.W.).

## Author contributions

Q.-F.L. conceived the project and designed the experiments. Q.-F.L. and Q.-Q.L. supervised the project, acquired funding and revised the manuscript. J.-D.W., J.W., L.-C.H., L.-J.K., C.-X.W., M.X., P.Z., and L.-H.Z. performed the experiments. J.-D.W., J.W., L.-C.H., Q.-Q.L. and Q.-F.L. analyzed the data. C.C., D.-S.Z., X.-L.F., C.-Q.Z., Y.Z., and L.Z. participated in data analysis and coordination of the study. J.-D.W. and Q.-F.L. wrote the manuscript. All authors read and approved the final version of the manuscript.

## Competing interests

The authors declare no competing interests.
