## [Peer Review File · Nature Communications]

REVIEWER COMMENTS

Reviewer #1 (Remarks to the Author):

In this manuscript, Wang et al. reported an ABA-responsive molecular module NF-YB1–SLRL2–bHLH144 that functions in the ABA-mediated regulation of rice grain quality and pre-harvested sprouting resistance. The work seems quite innovative, especially in the demonstration that a member of the rice DELLA family, SLRL2, has DNA binding activity, and its function is not related to GA but to ABA. The authors had done a lot of work. Some experiments were well designed and conducted, and the results were nicely organized and presented. Unfortunately, the authors seem to lack in-depth thinking on the intrinsic connections between some complex results, and the overall logical structure of this manuscript leaves something to be desired.

Major concerns:

1. The authors reported that ABA can repress the expression of the Wx gene and reduce AC. They discovered this important phenomenon by spraying 100 μ M ABA on rice panicles every two days from 5 DAF to 15 DAF. Due to the lack of a clear introduction to the relationship between ABA and AC, it is hard for readers outside the field to determine whether it is a new role of ABA discovered by authors. Anyway, the concentration of 100 μ M ABA is much higher than the normal physiological one in seeds and may cause various ABA-related cross-talking and developmental events, which may play a role in Wx expression and amylose synthesis. Therefore, it is necessary to be cautious about whether the alteration in Wx expression and AC are truly regulated by ABA.

We suggest to supplement several experiments in order to support their conclusion:

A. Treat the sample using a series of ABA concentration gradients and check whether the degree of Wx and AC alteration is ABA dosage-dependent.

B. Check whether AC and Wx expression have corresponding changes in the mutants deficient in ABA biosynthesis/catabolism. If these genetic material were not prepared in advance, WT treated with ABA biosynthesis inhibitor can mimic the mutants deficient in ABA biosynthesis.

2. The authors claimed that ABA regulates rice grain quality and seed dormancy via the NF-YB1-SLR2-bHLH144 module. However, based on the limited experimental evidence provided by authors, it is hard to determine whether the regulation of ABA is dependent on the module or the regulation of module is dependent on ABA, even whether the two are in the same regulatory pathway remains unknown. Much more genetic data are needed to establish the relationship between ABA and this module in regulating AC and PHS.

3. The PHS-related phenotype in the genetic materials are much more pronounced than the rice quality-related ones (AC, GC, etc). However, the authors did not devote much effort to elucidating the molecular mechanism of the this module in regulating seed dormancy. There are obvious shortcomings in this part of the work compared to the work on “module-Wx-AC/GC-rice quality”. The dormancy part of work could be removed since they are not closely related to the major highlight of this article.

Minor points:

1. In this manuscript, the authors claimed that SLRL2, bHLH144, and NF-YB1 form a complex functional module that mediates the regulation of ABA on AC in rice by modulating the expression of Wx genes. In addition, the authors introduced that Wx is a major gene that regulates not only AC but also GC (line 49). Therefore, in support of their claim, the AC, GC, as well as the Wx expression are all necessary to be examined in all genetic materials and ABA treatment. The authors have already examined the effects of ABA on AC (Fig.1a), GC (Fig.S1f) and Wx (Fig.1b[qRT-PCR]); the effects of SLRL2 on AC (Fig.1j), GC (Fig. S6b) and Wx (Fig3b, i [LUC/Rluc] and Fig. S6a [qRT-PCR]); the effect of NF-YB1 on AC (Fig. S10a), GC (Fig. S10b) and Wx (Fig. 3c [ChIP]), 3e [EMSA] and 3h, i [LUC/Rluc]); and the effects of slrl2 nf-yb1 on AC (Fig. 4q). But the effect of NF-YB1, slrl2 nf-yb1, and slrl2 bhlh144 on Wx (qRT-PCR), the effect of SLRL2 on Wx (EMSA) and so on, are not determined. We suggest the authors provide the relevant results in the modified version to clarify these questions and put all of these result into Figures according to a suitable logic structure. Meanwhile, the results that are not closely related to the main discovery of this work (such as Fig. 1d-i) could be moved into the Supplementary Figures.
2. To investigate whether a gene is responsive to a phytohormone, a treatment time of less than 3 hours is optimal (for example, the GA treatment in Fig. S4b). However, the ABA treatment time is too long in Figure 1b and 1c.
3. In Fig. 4a,b, the RNA and protein should be extracted from panicles or seeds rather than 14-d seedlings.
4. In Fig. 2e, it seems that the protoplast (SLRS2-nYFP NF-YB1-cYFP) is not health because the fluorescent signal in the nucleus is weird.
5. In Figs. 4d and 4j, it seems that there are shifted probes in the Mutation lane, which should not exist.
6. In Fig. 4g, at least add another region besides P1 as a negative control.
7. In Figs. 4h and 4l, there is no indication of which tissue the RNA was extracted from.
8. In Fig. S9i-l, the nf-yb1 mutants showed reduced grain length and width, but the slrl2 mutants did not (Fig. S5). Since they form a regulatory module in regulating seed traits, please try to provide a reasonable discussion of their differences.

Reviewer #2 (Remarks to the Author):

The manuscript of Wang et al. reports an identification of an ABA regulated molecular mechanism, i.e. the NF-YB1-SLRL2-bHLH144 regulatory module, that play role in regulation of rice quality regulation via targeting the Wx gene. Due to NF-YB1 and bHLH144 module was reported before, the novelty including these two components was reduced.

ABA treatment of rice panicle at the filling stage induced a reduced 1000-grain weight of harvested rice seeds (Fig S1) and also reduced amylose content (Fig 1). It looks that grain weigh was reduced more than amylose content, why did the authors think the identified candidate transcription factor is involved in regulating rice quality rather than grain weight? Although ABA treatment may change the amylose content, this change is very small, and application of ABA to regulate amylose content and rice quality is impossible, only in the stressed conditions were the induced ABA possible to regulate the rice quality.

The NY-YB1-bHLH144 module to regulate Wx gene expression was already reported before by Bello et al. (2019), who indicated that NF-YB1-YC12-bHLH144 regulate grain quality in rice via regulating the Wx gene. Because the modules shared the NF-YB1 and bHLH144, in which condition the NF-YB1-YC12-bHLH144 works, and in which condition NF-YB1-SLRL2-bHLH144 works. Because both modules regulate Wx gene, whether it is possible the module including four components (NF-YB1-SLRL2-YC12-bHLH144) exist? It should be interesting to address this.

Wx gene is an important target in this study, so expression of Wx gene and Western blot of GBSS should be provided in the slrl2, SLRL2-OX, nf-yb1 and bhlh144 materials.

In Fig. 6: the model did not show that the SLRL2 works downsteam of NF-YB1 and bHLH144.

L263-264: this sentence is difficult to follow, what is the two sequences? In L272, it indicated that bHLH144 was included in the DAP-seq data, whether the two sequences including the bHLH144.

L53: What is soft rice, whether is it a scientific term that has been accepted in rice community.

Response to Reviewers' comments

REVIEWER COMMENTS

Reviewer #1 (Remarks to the Author):

In this manuscript, Wang et al. reported an ABA-responsive molecular module NF-YB1–SLRL2–bHLH144 that functions in the ABA-mediated regulation of rice grain quality and pre-harvested sprouting resistance. The work seems quite innovative, especially in the demonstration that a member of the rice DELLA family, SLRL2, has DNA binding activity, and its function is not related to GA but to ABA. The authors had done a lot of work. Some experiments were well designed and conducted, and the results were nicely organized and presented. Unfortunately, the authors seem to lack in-depth thinking on the intrinsic connections between some complex results, and the overall logical structure of this manuscript leaves something to be desired.

Response: Thank you very much for your positive and insightful comments. Based on your detailed comments and suggestions, we have conducted additional experiments and made extensive revisions. In addition, we have carefully addressed all your comments and concerns. With your help, we have further strengthened our data and significantly improved the quality of the paper. The detailed information can be found in the following responses.

Major concerns:

1. The authors reported that ABA can repress the expression of the Wx gene and reduce AC. They discovered this important phenomenon by spraying 100 μ M ABA on rice panicles every two days from 5 DAF to 15 DAF. Due to the lack of a clear introduction to the relationship between ABA and AC, it is hard for readers outside the field to determine whether it is a new role of ABA discovered by authors. Anyway, the

concentration of 100 μM ABA is much higher than the normal physiological one in seeds and may cause various ABA-related cross-talking and developmental events, which may play a role in Wx expression and amylose synthesis. Therefore, it is necessary to be cautious about whether the alteration in Wx expression and AC are truly regulated by ABA.

Response: Thank you for your expert comments and valuable suggestions.

(1) We agree with the reviewer expert that more introduction on the relationship between ABA and AC should be provided. In fact, a number of publications indicate that ABA is closely involved in the regulation of grain filling and endosperm development in rice. For example, ABA levels increased with seed development, reaching a peak around 6 days after flowering (DAF), and then gradually decreased until seed maturation (Zhang et al., 2020), suggesting its essential role in regulating grain filling and seed maturation. Physiological experiments showed that soil desiccation increased ABA content in rice spikelets, and ABA content was significantly correlated with grain filling rate and sink activities under moderate soil desiccation conditions (Wang et al., 2015). In a more recent study, Qin et al (2021) demonstrated that grain-filling 1 (DG1) can mediate long-distance ABA transport efficiency and its mutation caused abnormal grain filling characterized by aberrantly loose starch granules and reduced 1000-grain weight. Starch is the major component of rice seeds, accounting for about 90% of their dry weight, and starch biosynthesis is the main event of grain filling. Both ABA metabolism and transport are crucial for the biosynthesis of starch, which is composed of amylose and amylopectin. A more direct evidence from another physiological study in rice, Chen et al (2019) showed that ABA treatment (10 μM) along led to certain decrease of rice AC, similar our extent, but their focus was the combination treatment with ABA (10 μM) and sucrose (15 mM), which caused a more pronounced AC decrease. However, only physiological phenotypes were provided, without any progress on the molecular mechanism.

Taken together, previous studies have shown that ABA is important in regulating grain filling and starch biosynthesis in rice. Only one study showed some initial

physiological data that ABA (10 μM) negatively regulates rice AC, which is consistent with our result by using 100 μM ABA for treatment, providing some evidence to show no dosage effect of ABA on AC change.

Therefore, we added more information in the Introduction section of the revised manuscript. “In addition, ABA is also a positive regulator of rice grain filling by modulating starch biosynthesis¹⁸. The physiological evidence indicated that ABA treatment could slightly reduce rice AC, which has potential application in improving rice grain quality¹⁹. *NCED* genes control the limiting step of ABA biosynthesis in plants. For example, *OsNCED3* is highly expressed in developing seeds and is involved in the regulation of rice grain development and PHS²⁰. In addition, the *DGI* gene controls the transport of ABA from rice stems to seeds, and the filling rate of *dgl* mutant seeds was slower²¹. Although these evidences indicated that ABA is involved in the regulation of grain filling and rice quality, the precise mechanism by which ABA regulates rice quality has not been reported.”

(2) Thank you very much for your critical comments on the issue of ABA concentration. Yes, the ABA concentration (100 μM) used in our study seems relatively high. However, this was our main concern when we conducted the ABA treatment experiment. We chose this concentration of ABA mainly for the following reasons. Firstly, the endogenous ABA content in various organs of rice was measured by Qin et al (2021). The ABA content was approximately 2400 ng/g FW in caryopsis, 100 ng/g FW in stem and 150 ng/g FW in leaf. Based on these data and the molecular weight of ABA (264 g/mol), we calculated that the actual ABA concentration in caryopsis was about 9.1 μM , about 15 times higher than that in the leaf (about 0.57 μM). In the studies on ABA treatment of rice leaves, the commonly used concentration of ABA was about 5 to 10 μM (Li et al., 2021; Zhang et al., 2023), almost 10 times higher than the endogenous ABA content in the leaf. Second, we treated the rice panicle in the field by spraying ABA to avoid rice transplantation and keep its natural status. However, ABA solution is light sensitive and easily to be degraded by strong light. Also, the temperature is high and the light intensity is strong in the field of Yangzhou in summer.

In addition, the rice endosperm is covered by the seed hull. Therefore, we used a relatively high concentration of ABA to ensure the treatment effect on endosperm under natural conditions. Thirdly, we sprayed ABA on the main panicle of each rice plant at 5 DAF. At this stage, the panicle number and grain number per panicle were hardly affected by the ABA treatment. Local application of ABA to only one panicle can reduce the possibility of other ABA-related cross-talk or side effects on plant growth and development. This allows the effect of ABA on both starch biosynthesis during endosperm development and the final amylose content (AC) in the mature seeds to be studied.

References:

- Feng, T. et al. OsMADS14 and NF-YB1 cooperate in the direct activation of *OsAGPL2* and *Wx* during starch synthesis in rice endosperm. *New Phytol.* **234**, 77–92 (2022)
- Xiong, Y. et al. NF-YC12 is a key multi-functional regulator of accumulation of seed storage substances in rice. *J. Exp. Bot.* **70**(15), 3765–3780 (2019)
- Xu, J. J., Zhang, X. F., & Xue, H. W. Rice aleurone layer specific OsNF-YB1 regulates grain filling and endosperm development by interacting with an ERF transcription factor. *J. Exp. Bot.* **67**(22), 6399–6411 (2016)
- Qin, P. et al. Leaf-derived ABA regulates rice seed development via a transporter-mediated and temperature-sensitive mechanism. *Sci. Adv.* **7**, eabc8873 (2021)
- Zhang, X. F. et al. Phytohormone dynamics in developing endosperm influence rice grain shape and quality. *J Integr Plant Biol.* **62**(10), 1625–1637 (2020)
- Wang, Z. et al. Abscisic acid and the key enzymes and genes in sucrose-to-starch conversion in rice spikelets in response to soil drying during grain filling. *Planta.* **241**(5), 1091–1107 (2015)
- Chen, T. et al. Abscisic acid synergizes with sucrose to enhance grain yield and quality of rice by improving the source-sink relationship. *BMC Plant Biol.* **19**(1), 525 (2019)
- Li, Q. et al. Synergistic interplay of ABA and BR signal in regulating plant growth and

adaptation. *Nat plants*. 7(8), 1108–1118(2021)

Zhang, Q. et al. Stomatal conductance in rice leaves and panicles responds differently to abscisic acid and soil drought. *J Exp Bot*. 74(5), 1551–1563 (2023)

We suggest to supplement several experiments in order to support their conclusion:

A. Treat the sample using a series of ABA concentration gradients and check whether the degree of *Wx* and AC alteration is ABA dosage-dependent.

B. Check whether AC and *Wx* expression have corresponding changes in the mutants deficient in ABA biosynthesis/catabolism. If these genetic material were not prepared in advance, WT treated with ABA biosynthesis inhibitor can mimic the mutants deficient in ABA biosynthesis.

Response: Thank you very much for your insightful comments and professional suggestions. Following your suggestions, we have carried out further experiments to confirm that the change in *Wx* expression and AC is indeed regulated by ABA.

(1) Response to the suggestion A. At present, we don't have rice panicles for ABA treatment. Therefore, we used rice seedlings and isolated protoplasts to study the expression changes of *SLRL2* and *Wx* in response to ABA. First, rice seedlings were subjected to short-term ABA treatment. The result showed that only 30 min of ABA treatment significantly promoted the expression of *SLRL2* and suppressed the expression of *Wx* (Response Figure 1-1 a, b). Secondly, the dual luciferase system was used to check whether the change in expression of *Wx* and *SLRL2* was ABA dose-dependent. Three different concentrations of ABA (10 μ M, 20 μ M, 100 μ M) were used. The result showed that all three different concentrations of ABA induced a similar degree of change in the expression of both *SLRL2* and *Wx*. More specifically, ABA induced the expression of *SLRL2* and suppressed the expression of *Wx* in a dose-independent manner (Response Figure 1-1 c, d). Regarding the relationship between *Wx* expression and AC, it is well known that *Wx* expression has a strong positive

correlation with rice AC (Huang et al., 2020; Zeng et al., 2020). Therefore, we concluded that the degree of *SLRL2*, *Wx* and AC alteration was not ABA dosage-dependent.

(2) Response to the suggestion B. Because one of our research interests is to dissect the regulatory mechanism of rice seed germination and dormancy. Some genetic material related to ABA metabolism and transport has already been generated by Chu-Xin Wang and Min Xiong, two members of our laboratory. *NECD2* is one of the key enzymes involved in ABA biosynthesis, while *DG1* is a critical ABA transporter for long-distance ABA transport. Fortunately, both the *nced2* and *dgl* mutants were homozygous and their quality traits were analysed (Response Figure 1-1 e-h). GC was decreased, whereas both AC and *Wx* expression were increased in *nced2* and *dgl* mutants. These data suggest that ABA negatively regulates *Wx* expression and rice AC (Response Figure 1-1 g-i).

Fluridone, an ABA biosynthesis inhibitor, was also used to treat rice panicles in summer to mimic the ABA-deficient condition. Because the fluridone treatment resulted in severe preharvest sprouting (PHS), we didn't analyse the physiochemical properties of the seeds in the initial submission. During the revision phase, we carefully selected the ungerminated seeds from the fluridone-treated samples and analysed AC and GC. The data showed that rice AC was increased in response to fluridone treatment, exactly the opposite of ABA (Response Figure 1-1 j, k). Gene expression analysis showed that the expression of *SLRL2* was decreased while that of *Wx* was increased in fluridone-treated developing seeds (Response Figure 1-1 l, m).

All the above evidence showed that ABA does indeed negatively regulate *Wx* expression and rice AC. We have included all the new data and the description in the revised manuscript. In addition, Chu-Xin Wang and Min Xiong have been added to the list of co-authors in the revised manuscript for their contributions to the ABA-related genetic material and some experiments.

Response Figure 1-1. ABA negatively regulates *Wx* expression and rice AC through *SLRL2*. *Wx* (a), *SLRL2* (b) and *NF-YC9* (b) expression under short-term ABA treatment. The 14-d-old rice seedlings were transferred to a medium containing 50 μ M ABA, and rice samples were collected at 0, 30 and 60 min after ABA treatment, respectively. *OsUBC* was used as an internal control to normalize gene expression. The expression level of *Wx* was set to 1 for samples without ABA treatment. Error bars represent SD (n = 3 biological replicates). Dual luciferase reporter assay to study the regulation of *Wx* (c) and *SLRL2* (d) gene expression to ABA treatment. Fluorescence ratio was measured after treatment with a series of ABA concentration gradients (0, 10, 50 and 100 μ M) for 30 min. Error bars represent SD (n = 5 biological replicates). Schematic diagram of the gene structure of *NCED2* (e) and *DG1* (f), and their respective CRISPR/Cas9 gene editing target information. g AC of *nced2* and *dg1* mutants. h GC of the *nced2* and *dg1* mutants. i Expression of *Wx* in the mature seeds

of *nced2* and *dg1* mutants. Rice AC (**j**) and GC (**k**) in response to ABA or fluridone treatment. Expression of *Wx* (**l**), *SLRL2* (**m**) and *NF-YC9* (**n**) in response to ABA or fluridone treatment. 100 μ M ABA or 100 μ M fluridone was sprayed every two days on rice panicles at 5 days after flowering (DAF), and grain samples were collected at 15 DAF. In (**g-m**), error bars represent SD (n = 3 biological replicates). Different lowercase letters above the error bars indicate significant differences according to a one-way ANOVA with Tukey's test ($P < 0.05$).

2. The authors claimed that ABA regulates rice grain quality and seed dormancy via the NF-YB1-SLR2-bHLH144 module. However, based on the limited experimental evidence provided by authors, it is hard to determine whether the regulation of ABA is dependent on the module or the regulation of module is dependent on ABA, even whether the two are in the same regulatory pathway remains unknown. Much more genetic data are needed to establish the relationship between ABA and this module in regulating AC and PHS.

Response: Thank you for your critical comments and suggestions. Our original data and newly added evidence indicated that the regulation of the NF-YB1-SLRL2-bHLH144 module is dependent on ABA. First, *SLRL2*, a key component of the module, was induced by ABA and repressed by fluridone in rice panicles (Response Figure 1-2a). Second, both short-term ABA treatment and treating of protoplasts using a series of ABA concentration gradients showed that the expression of *SLRL2* was induced by ABA (Response Figure 1-1b, d). These data indicated that *SLRL2* was involved in the ABA pathway rather than GA. Thirdly, the *Wx* gene, as the crucial determinant of rice AC, is the direct target of *SLRL2* and its expression was suppressed by *SLRL2* (Response Figure 1-2b-f). In addition, the expression change of *Wx* to both ABA and fluridone treatment was closely correlated with the expression change of *SLRL2* (Response Figure 1-1a-d, l and m). Finally, both knockout and overexpression of *SLRL2* attenuated the sensitivity of *Wx* expression and rice AC to ABA treatment (Response Figure 1-2 b, g and h). Taken together, *SLRL2* functions as a key node between the

ABA signaling pathway and the NF-YB1-SLRL2-bHLH144 module, and its downstream target genes, including *Wx*.

The regulation of PHS is similar to that of AC. That is SLRL2 plays an important role in mediating the regulation of rice PHS by ABA. The mutation or overexpression of *SLRL2* attenuated the sensitivity of rice to ABA treatment by analysing the PHS phenotype (Response Figure 1-2i). As the molecular mechanism of this module in the regulation of seed dormancy was preliminary in the first submission, further experiments were designed and performed during the revision phase. Finally, *MFT2*, a key regulator of seed dormancy and PHS in rice, was identified as the direct target of SLRL2. As mentioned above, SLRL2 is the link between ABA and the regulatory module, we also successfully established the relationship between ABA and this module in regulating PHS. The detailed description of the MFT2 related results will be provided in the next response. Thank you very much.

Response Figure 1-2. SLRL2 mediates ABA regulation of AC and PHS in rice. **a** Expression of *SLRL2* and *NF-YC9* in response to ABA or fluridone treatment. 100 μ M ABA or 100 μ M fluridone was sprayed every two days on rice panicles at 5 days after flowering (DAF), and grain samples were collected at 15 DAF. **b** *Wx* gene expression levels in *SLRL2*-related materials. *OsActin01* was used as an internal control to normalize gene expression. The transcript abundance of *Wx* in the WT was set to 1. Error bars represent the SD (n = 3 biological replicates). **c** Dual-luciferase reporter assay to study the regulation of *Wx* gene expression by *SLRL2* and *NF-YB1*. The value from the 62-SK and *Wx* pro-LUC co-transformation group was set to 1. Error bars represent the SD (n = 5 biological replicates). **d** Schematic representation of the *Wx* promoter highlighting the G-box motif regions. The marked regions P1 to P3 were the positions used for the ChIP-qPCR assay. **e** ChIP-qPCR assay showed that the P1 region of the *Wx* promoter was also significantly enriched by *SLRL2*. Error bars represent SD (n = 3 experiments). Enriched values are normalized to the input. An unrelated *OsActin1* intron region was used as a negative control. IgG-immunoprecipitated DNA was used as control. ** P < 0.01 (Student's t-test). **f** EMSA experiments showed that *SLRL2* directly binds to the *Wx* promoter. **g** AC of *SLRL2* knockout and overexpression materials in response to ABA treatment. **h** *Wx* expression in *SLRL2* knockout and

overexpression materials in response to ABA treatment. In **g** and **h**, error bars represent the SD (n = 3 biological replicates). **i** Field PHS data of *SLRL2*-related materials in response to ABA treatment. Error bars represent SD (n = 5 biological replicates). Different lowercase letters above the error bars indicate significant differences according to a one-way ANOVA with Tukey's test ($P < 0.05$).

3. The PHS-related phenotype in the genetic materials are much more pronounced than the rice quality-related ones (AC, GC, etc). However, the authors did not devote much effort to elucidating the molecular mechanism of the this module in regulating seed dormancy. There are obvious shortcomings in this part of the work compared to the work on “module-*Wx*-AC/GC-rice quality”. The dormancy part of work could be removed since they are not closely related to the major highlight of this article.

Response: Thank you for your critical comments. Eating and cooking quality (ECQ) of rice is the most important consideration for consumers when choosing rice in the market, while AC is the most important parameter that determines the ECQ of rice. Fine-tuning AC is critical in the practice of elite rice breeding. For example, the rare *Wx* allele *Wx^{mw}*/*Wx^{la}* was identified by genome-wide association analysis and map-based cloning (Zhang et al., 2021; Zhou et al., 2021). Rice with the *Wx^{mw}*/*Wx^{la}* allele had an AC of about 14%, but its ECQ was much better than that of rice with the *Wx^b* allele, whose AC was about 16%. Thus, fine-tuning the AC is an important strategy for improving ECQ in rice. In addition, the original purpose of this project was to identify new regulators of *Wx* and AC. This is the main reason why, in the original manuscript, we paid more attention to investigating how the NF-YB1-SLRL2-bHLH144 module regulates AC.

However, as mentioned by the reviewer expert, the PHS-related phenotype is much more pronounced in the genetic material and is also a very important part of this manuscript. However, the data to reveal its molecular mechanism is not sufficient in the first submission. Therefore, we further analysed the DAP-seq data and identified *MFT2* as an important direct target of *SLRL2* mediating its regulation on seed dormancy.

MFT2 is a negative regulator of seed germination. *MFT2* knockout promoted pre-harvest sprouting in rice, whereas *MFT2* overexpression delayed seed germination (Song et al., 2020). A series of experiments, including qRT-PCR, dual luciferase assay, yeast one-hybrid, EMSA and CHIP-qPCR analysis, were carried out and showed that SLRL2 indeed binds directly to the promoter of *MFT2* and activates its expression (Response Figure 1-3a-f). We also benefited from the contribution of Chu-Xin Wang and Min Xiong, who had already generated the *mft2* mutant and analysed its PHS phenotype in the summer. The data showed that the *MFT2* mutation significantly promoted rice PHS (Response Figure 1-3 h, i), in agreement with the previous publication (Song et al., 2020). We believe that these new results have further improved the molecular mechanism of SLRL2 involved in the regulation of rice PHS. We have included all these data in the new version of the manuscript and revised our manuscript accordingly.

Response Figure 1-3. SLRL2 directly activates the transcription of *MFT2* to inhibit rice PHS. **a** Expression analysis of *MFT2* in *slr2* mutant and SLRL2 overexpression transgenic rice. *OsActin01* was used as an internal control to normalize gene expression. The expression level of *MFT2* in WT was set to 1. Error bars represent SD (n = 3 biological replicates). **b** Dual-luciferase reporter assay showed that SLRL2 activates the expression of *MFT2*. The value of the 62-SK and *MFT2* pro-LUC co-transfection group was set to 1. Error bars represent SD (n = 5 biological replicates). **c**

Structural diagram of the SLRL2-binding cis-acting element on the *MFT2* promoter. **d** Yeast one-hybrid experiments showing the direct interaction between SLRL2 and the *MFT2* promoter. **e** EMSA experiment confirming the direct interaction between SLRL2 and the *MFT2* promoter. The hot probe is a biotin-labeled *MFT2* promoter fragment with a CTCC motif, while the competition probe is unlabeled. **f** The ChIP-qPCR assay shows the enrichment of the *MFT2* promoter by SLRL2 in vivo. Error bars represent SD (n = 3 experiments). In **b** and **f**, ** P < 0.01 (Student's t-test). **g** Schematic diagram of the *MFT2* gene structure and corresponding CRISPR/Cas9 gene editing target information. Panicle phenotypes (**h**) and quantification curve (**i**) of PHS analysis of the *mft2* mutant and its WT. The scale bar is 1 cm. Error bars represent SD (n = 5 biological replicates).

References:

Zhang, C. et al. A rare *Waxy* allele coordinately improves rice eating and cooking quality and grain transparency. *J Integr Plant Biol.* **63**(5), 889–901 (2021)

Zhou, H. et al. The origin of *Wx^{la}* provides new insights into the improvement of grain quality in rice. *J Integr Plant Biol.* **63**(5), 878–888 (2021)

Song, S. et al. OsMFT2 is involved in the regulation of ABA signaling-mediated seed germination through interacting with OsbZIP23/66/72 in rice. *Plant J.* **103**(2), 532–546 (2020)

Minor points:

1. In this manuscript, the authors claimed that SLRL2, bHLH144, and NF-YB1 form a complex functional module that mediates the regulation of ABA on AC in rice by modulating the expression of Wx genes. In addition, the authors introduced that Wx is a major gene that regulates not only AC but also GC (line 49). Therefore, in support of their claim, the AC, GC, as well as the Wx expression are all necessary to be examined in all genetic materials and ABA treatment. The authors have already examined the effects of ABA on AC (Fig.1a), GC (Fig.S1f) and Wx (Fig.1b[qRT-PCR]); the effects of SLRL2 on AC (Fig.1j), GC (Fig. S6b) and Wx (Fig3b, i [LUC/Rluc] and Fig. S6a

[qRT-PCR]); the effect of NF-YB1 on AC (Fig. S10a), GC (Fig. S10b) and Wx (Fig. 3c [ChIP]), 3e [EMSA] and 3h, i [LUC/Rluc]); and the effects of *slrl2 nf-yb1* on AC (Fig. 4q). But the effect of NF-YB1, *slrl2 nf-yb1*, and *slrl2 bhlh144* on Wx (qRT-PCR), the effect of SLRL2 on Wx (EMSA) and so on, are not determined. We suggest the authors provide the relevant results in the modified version to clarify these questions and put all of these result into Figures according to a suitable logic structure. Meanwhile, the results that are not closely related to the main discovery of this work (such as Fig. 1d-i) could be moved into the Supplementary Figures.

Response: Thank you very much for your detailed comments and valuable suggestions. Following your suggestion, we have performed the experiments and found that the mutation of either *NF-YB1* or *bHLH144* reduced *Wx* expression (Response Figure 1-4a). In addition, the expression of *Wx* was significantly increased in both *slrl2 nf-yb1* and *slrl2 bhlh144* double mutants, similar to the *slrl2* mutant (Response Figure 1-4b). Regarding the effect of SLRL2 on *Wx* (EMSA), we have provided the EMSA result in the original Fig. 4d. The reason we chose to show the data there is that we had previously thought that SLRL2, like SLR1, didn't have direct DNA-binding activity. However, after the yeast one-hybrid and DAP-seq analysis, we found that SLRL2 is a transcription factor with DNA binding activity. Therefore, EMSA analysis was performed and confirmed that SLRL2 can also bind directly to the promoter of *Wx*. We apologise for the improper organization of the figures. Here we have moved this EMSA result to the Supplementary Fig. 9 (Response Figure 1-5) where it is easier to see.

We have added all these results to the figures according to an appropriate logical structure. We have also moved the results that are not closely related to the main discovery of this work into the Supplementary Figures. For example, the original Fig. 1d-i has been moved to Supplementary Fig. 2e-i, Supplementary Fig. 4d and Supplementary Fig. 5a,c.

Response Figure 1-4 Analysis of *Wx* expression in single and double mutants of *slrl2*, *nf-yb1* and *bHLH144*. **a**, Transcription levels of *Wx* genes in *nf-yb1* and *bhlh144* mutants. **b**, Expression levels of *Wx* genes in single and double rice mutants concerning *SLRL2*, *NF-YB1*, and *bHLH144* genes. Set *Wx* gene expression in WT to 1. Error bars represent SD (n = 3 biological replicates). Different lowercase letters above error bars indicate significant differences analyzed by one-way ANOVA with Tukey's post-hoc test.

Response Figure 1-5 EMSA experiments showed the direct binding of SLRL2 to the *Wx* promoter.

2. To investigate whether a gene is responsive to a phytohormone, a treatment time of less than 3 hours is optimal (for example, the GA treatment in Fig. S4b). However, the ABA treatment time is too long in Figure 1b and 1c.

Response: Thank you for your comments. Yes, I agree with the reviewer expert that a treatment time of less than 3 hours is optimal for investigating whether a gene responds

to a phytohormone. However, at the time we conducted the experiment, we treated the rice panicles in the field under high temperature and strong light. In addition, the rice endosperm is covered by the seed hull. Therefore, we treated the panicles every two days to ensure the effect of ABA on starch biosynthesis in the endosperm. Seed samples for expression analysis were collected at 15 DAF when the *Wx* gene had the highest expression level (Response Figure 1-6). We collected the seed samples only about 2 hours after the ABA treatment on that day. We reasoned that the effect of sprayed ABA applied two days earlier (13 DAF) on *Wx* transcription should be small under natural field conditions. Therefore, we used the samples to test the expression response of *Wx*, *SLRL2* and *NF-YC9* to ABA.

To be more precise, we also used the short-term treatment method to confirm the result, following your suggestion. Here, 30 min and 60 min ABA treatments were used, and the result showed that even 30 min ABA treatment remarkably promoted the expression of *SLRL2* and suppressed the expression of *Wx* (Response Figure 1-1a, b), suggesting that these two genes are quite sensitive to ABA. As for *NF-YC9*, no change in expression was observed even after 60 min of treatment (Response Figure 1-1b). We also used the dual luciferase system to confirm this result. By using three different concentrations of ABA (10 μ M, 20 μ M, 100 μ M) for a short time treatment (30 min), the result showed that the expression of *Wx* was significantly reduced, while that of *SLRL2* was significantly increased in an ABA dosage-independent manner (Response Figure 1-1 c and d).

Response Figure 1-6 Spatio-temporal expression patterns of *Wx*. *OsActin01* was used as an internal control for normalization of gene expression data. The expression level of *Wx* in seeds five days after flowering (DAF) was set to 1. Young root (YR) and

young shoot (YS) samples were collected from 14-day-old rice seedlings. Stem, leaf blade, leaf sheath, and young panicle (YP) samples were all from rice at the booting stage. Developing seed samples were collected at 0, 5, 10, 15, 20, 25, and 30 DAF. Error bars represent SD (n = 3 biological replicates).

3. In Fig. 4a,b, the RNA and protein should be extracted from panicles or seeds rather than 14-d seedlings.

Response: Thanks for the helpful comments. Yes, it's better to extract the RNA and protein from rice panicles for expression analysis. However, we don't have rice panicles for experiments at present. Since GA had no effect on *SLRL2* transcription (Itoh et al., 2005), we only used GA to treat the seedlings to confirm this result. In fact, our RT-qPCR result was consistent with the previous publication. We then further investigated the expression of *SLRL2* in response to GA by using the dual luciferase system. The data also showed that GA had no effect on *SLRL2* expression (Response Figure 1-7a), but the expression of *SLRL2* was significantly promoted by ABA in the same system (Response Figure 1-1d).

In terms of protein analysis, since the *SLRL2* gene is driven by the Actin promoter, it had a constitutive expression pattern in rice plants. Therefore, either panicle or leaf organs could be used to test how GA affects *SLRL2* at the protein level. The WB result indicated that GA didn't affect the stability of the *SLRL2* protein in rice (Response Figure 1-7b). To further confirm this notion, we also expressed the *SLRL2*-GFP protein in rice protoplasts for GA treatment. The WB result showed that no significant difference could be observed between the samples with or without GA treatment (Response Figure 1-7c), consistent with that in rice seedlings.

Response Figure 1-7 *SLRL2* is not sensitive to GA. **a** Dual luciferase reporter assay was used to investigate the response of *SLRL2* to GA treatment. The fluorescence ratio was measured after treatment with 50 μ M GA for 30 min. **b** Analysis of *SLRL2* protein in response to GA treatment. The seedlings of transgenic rice overexpressing *SLRL2* were used for GA treatment and subsequent Western blot analysis. Hsp82 was used as an internal control to normalize the protein loading. *SLRL2*-Flag and HSP82 proteins were detected by using anti-Flag and anti-HSP82 antibodies, respectively. **c** The abundance of *SLRL2* protein in response to GA treatment. Hsp82 was used as an internal control to normalize the protein loading. The *SLRL2*-GFP and HSP82 proteins were detected by using anti-GFP and anti-HSP82 antibodies, respectively.

4. In Fig. 2e, it seems that the protoplast (*SLRS2*-nYFP NF-YB1-cYFP) is not healthy because the fluorescent signal in the nucleus is weird.

Response: Yes, we have redone the BiFC analysis by using more healthy protoplasts. The original photo has been replaced by the new one in the revised manuscript (Response Figure 1-8). Thank you very much.

Response Figure 1-8 Bimolecular fluorescence complementation (BiFC) analysis revealed the nuclear interaction between SLRL2 and NF-YB1 in rice protoplasts. Two sets of plasmid groups, including SLRL2-nYFP and cYFP, nYFP and NF-YB1-cYFP, were used as negative controls. The scale bar is 10uM.

5. In Figs. 4d and 4j, it seems that there are shifted probes in the Mutation lane, which should not exist.

Response: We thank the reviewer expert for the comments. In fact, the mutation lane here means that we used the unlabelled mutation probe (50×) to compete with the labelled probe. Therefore, the last mutation lane still shows clearly shifted probes compared to the lane just before it, where the 50× unlabelled normal competitor was added. We apologize for the unclear labelling and annotation in the original submission. We have improved this in the revised manuscript (Response Figure 1-5 and 1-9).

Response Figure 1-9 EMSA experiments demonstrated that SLRL2 can directly bind *bHLH144* promoter.

6. In Fig. 4g, at least add another region besides P1 as a negative control.

Response: Yes, following your suggestion, we have added P2, another region besides P1 as a negative control (Response Figure 1-10), which provides stronger support to show that SLRL2 binds specifically to the *bHLH144* promoter. Thank you very much.

Response Figure 1-10 CHIP-PCR experiments showed that SLRL2 binds to the *bHLH144* promoter in vivo. a Schematic representation of the *bHLH144* promoter and the potential SLRL2 binding site. **b** ChIP-qPCR assay showed that the P1 region of the *bHLH144* promoter was significantly enriched by SLRL2. Enriched values are normalized to the input. IgG-immunoprecipitated DNA was used as the control. Error bars represent SD (n = 3 experiments). ** P < 0.01 (Student's t-test).

7. In Figs. 4h and 4l, there is no indication of which tissue the RNA was extracted from.

Response: Thank you for your helpful comments. The RNA was extracted from the developing seeds at 15 DAF. We have added this information in the revised manuscript.

8. In Fig. S9i-l, the *nf-yb1* mutants showed reduced grain length and width, but the *slrl2* mutants did not (Fig. S5). Since they form a regulatory module in regulating seed traits, please try to provide a reasonable discussion of their differences.

Response: Thank you for your insightful comments. Yes, the *nf-yb1* mutants showed reduced grain length and width (Fig. S9i-l), which is consistent with previous publications (Bello et al., 2019). With respect to *slrl2*, no changes in grain size were observed. Although NF-YB1 and SLRL2 formed a regulatory module, they mainly co-regulated rice AC and seed dormancy, but not seed size. There are two possible explanations for this difference. First, NF-YB1 also interacts with several other transcription factors, such as NF-YC12, OsERF115 and OsMADS14 (Xiong et al., 2019; Xu et al., 2016; Feng et al., 2022). Some of these NF-YB1 interacting proteins play important roles in controlling grain size. However, SLRL2 is an uncharacterised new transcription factor, there are no publications reporting its interacting proteins. The different interaction partners of NF-YB1 and SLRL2 may be an important reason for their different roles in regulating seed traits. Secondly, the spatial and temporal difference in the expression pattern of *NF-YB1* and *SLRL2* may be another important reason. *SLRL2* is dominantly expressed in seeds, but not in young panicles or other tissues tested. Although the expression pattern of NF-YB1 and SLRL2 was similar in developing seeds, NF-YB1 still had some low expression in other tissues, including young panicles. It is well known that the size of the seed hull is the main determinant of rice grain size. Therefore, it is reasonable that both NF-YB1 and SLRL2 regulate rice AC and seed dormancy, but only NF-YB1 controls grain size.

References:

Bello, B.K. et al. NF-YB1-YC12-bHLH144 complex directly activates *Wx* to regulate grain quality in rice (*Oryza sativa L.*). *Plant Biotechnol J.* **17**, 1222-1235 (2019)

Xiong, Y. et al. NF-YC12 is a key multi-functional regulator of accumulation of seed storage substances in rice. *J. Exp. Bot.* **70**(15), 3765–3780 (2019)

Xu, J. J., Zhang, X. F., & Xue, H. W. Rice aleurone layer specific OsNF-YB1 regulates grain filling and endosperm development by interacting with an ERF transcription factor. *J. Exp. Bot.* **67**(22), 6399–6411 (2016)

Feng, T. et al. OsMADS14 and NF-YB1 cooperate in the direct activation of OsAGPL2 and *Wx* during starch synthesis in rice endosperm. *New Phytol.* **234**, 77–92 (2022)

Thank you again for all your insightful comments and valuable suggestions.

Reviewer #2 (Remarks to the Author):

The manuscript of Wang et al. reports an identification of an ABA regulated molecular mechanism, i.e. the NF-YB1-SLRL2-bHLH144 regulatory module, that play role in regulation of rice quality regulation via targeting the *Wx* gene. Due to NF-YB1 and bHLH144 module was reported before, the novelty including these two components was reduced.

Response: We appreciate your kind interest and positive comments. The main finding of this study is the successful identification of a new transcription factor, SLRL2, and the demonstration of a SLRL2-centered regulatory module in the regulation of both rice

AC and seed dormancy, which is different from the reported NF-YB1 and bHLH144 module. More specifically, the novelties of our study include the following aspects. First, as a DELLA family member, SLRL2 is induced by ABA but is insensitive to GA. Second, SLRL2 had direct DNA-binding activity and its target gene network was revealed, which is quite different from SLR1. Third, we successfully identified and dissected several key targets of SLRL2, including *Wx*, *bHLH144* and *MFT2*, to mediate its regulation of rice AC and seed dormancy. Fourth, SLRL2 directly interacts with NF-YB1 protein to co-regulate the downstream target genes and seed-related traits, not only rice AC but also seed dormancy. Fifth, SLRL2 competes with NF-YC12 to interact with NF-YB1, thereby inhibiting the formation of the NF-YB1-YC12-bHLH144 complex, and consequently destabilising the NF-YB1 protein and reducing AC in rice. Therefore, the above information, together with a number of new findings from molecular analysis and rice genetic material during the revision stage, further improved the novelty and integrality of our manuscript. Thank you very much.

ABA treatment of rice panicle at the filling stage induced a reduced 1000-grain weight of harvested rice seeds (Fig S1) and also reduced amylose content (Fig 1). It looks that grain weight was reduced more than amylose content, why did the authors think the identified candidate transcription factor is involved in regulating rice quality rather than grain weight? Although ABA treatment may change the amylose content, this change is very small, and application of ABA to regulate amylose content and rice quality is impossible, only in the stressed conditions were the induced ABA possible to regulate the rice quality.

Response: Thank you for your insightful comments. We agree with the reviewer expert that the grain weight was reduced more than amylose content (AC) in response to ABA treatment. The reasons why we focused on studying SLRL2, which regulates rice quality rather than grain weight, are as follows. First, ABA and fluridone treatment

resulted in changes in both rice AC and grain weight (Response Figure 2-1a, b). Specifically, the change in grain weight was due to the change in grain thickness, which is consistent with the fact that ABA is an important hormone affecting rice grain filling. Secondly, the progress in research and breeding practice of rice quality, especially rice ECQ, lags far behind that of rice yield. Since AC is the most important parameter determining rice ECQ, the study of its regulatory molecular mechanism is quite important and necessary. In fact, fine-tuning AC is critical in elite rice breeding practice. For example, the rare *Wx* allele *Wx^{mw} / Wx^{la}* was identified by genome-wide association analysis and map-based cloning (Zhang et al., 2020b; Zhou et al., 2020b). Rice with the *Wx^{mw} / Wx^{la}* allele had an AC of about 14%, but its ECQ was much better than that of rice with the *Wx^b* allele, whose AC was about 16%. Third, it is interesting to note that knockout or overexpression of *SLRL2* had no effect on grain size and rice yield (Response Figure 2-1c-f), suggesting that *SLRL2* is an important element to specifically mediate ABA to regulate rice ECQ and seed dormancy.

The newly revealed module on ABA-regulated AC and seed dormancy is indeed very important. On the one hand, ABA is a critical plant hormone during rice seed development, and the establishment of this molecular module helps the reader to have a better understanding of the biological significance of ABA during seed development; on the other hand, various stresses are likely to reduce both the yield and quality of rice. However, we found that overexpression of *SLRL2* not only significantly increased the plant's resistance to PHS, thus preventing yield loss, but also reduced AC in rice, thus improving rice quality. Indeed, rice is often exposed to various stresses throughout its life cycle, in particular high temperature and high humidity. ABA is a critical stress-responsive plant hormone, and the frequent stress will induce the accumulation of ABA, thus triggering the activation of this molecular module. Therefore, our study not only dissected the novel molecular network centred on *SLRL2* in the regulation of AC and seed dormancy in rice, but also provided valuable genetic resources and genetic material for breeding elite rice with both superior ECQ and promoted PHS resistance. Even under natural stress conditions in the field, the *SLRL2*-centred mechanism may be helpful in improving rice quality and ensuring rice yield.

Response Figure 2-1. ABA or fluridone treatment, and yield traits of *SLRL2* related materials. **a** Rice AC in response to ABA or fluridone treatment. **b** 1000-grain weight in response to ABA or fluridone treatment. In **a** and **b**, error bars represent the SD ($n = 3$ biological replicates). Different lowercase letters above the error bars indicate significant differences according to a one-way ANOVA with Tukey's test ($P < 0.05$). Grain length (**c**), Grain width (**d**), 1000 grain weight (**e**) and Grain yield per plant (**f**) of *SLRL2*-related rice materials. Error bars represent SD ($n \geq 3$ biological replicates).

The NY-YB1-bHLH144 module to regulate *Wx* gene expression was already reported before by Bello et al. (2019), who indicated that NF-YB1-YC12-bHLH144 regulate grain quality in rice via regulating the *Wx* gene. Because the modules shared the NF-YB1 and bHLH144, in which condition the NF-YB1-YC12-bHLH144 works, and in which condition NF-YB1-*SLRL2*-bHLH144 works. Because both modules regulate *Wx* gene, whether it is possible the module including four components (NF-YB1-*SLRL2*-YC12-bHLH144) exist? It should be interesting to address this.

Response: Thank you for your critical comments and constructive suggestions. To

address your concerns, several experiments were designed and performed. First, we investigated which domain(s) of NF-YB1 mediate its interaction with two different transcription factors, SLRL2 and NF-YC12. The by Y2H result indicated that both SLRL2 and NF-YC12 interact with the same region of NF-YB1 (Response Figure 2-2a-c). Second, we sought to understand the biological effects of two different transcription factors binding to the same region of a protein. A luciferase complementation imaging (LCI) assay showed that SLRL2 inhibited the in vivo interaction between NF-YC12 and NF-YB1 (Response Figure 2-2d).

To further confirm this result, the cell-free degradation assay was performed (Response Figure 2-2e). Since it is known that the binding of NF-YC12 to NF-YB1 is pre-required for the formation of the NF-YB1-YC12-bHLH144 heterotrimer complex and thus for the stabilization of the NF-YB1 protein, we sought to determine whether SLRL2 enhances or attenuates the formation of the NF-YB1-YC12-bHLH144 complex by examining the stability of the NF-YB1 protein. The result showed that either the involvement of SLRL2 or NF-YC12 alone, or both proteins together, can stabilise NF-YB1. When bHLH144 was added, the NF-YB1 protein in the sample group with NF-YC12 (the sixth row) was more stable than that with SLRL2 (the fifth row), indicating that the formation of the NF-YB1-YC12-bHLH144 complex can indeed stabilise NF-YB1. Furthermore, when the four proteins, including NF-YB1, NF-YC12, bHLH144 and SLRL2, were all present in the same sample group, the NF-YB1 almost disappeared in the 60 min incubation lane (the seventh row), suggesting that SLRL2 competes with NF-YC12 to bind to NF-YB1, thereby affecting the formation of the NF-YB1-YC12-bHLH144 complex. To further confirm this notion, the mutant SLRL2 protein (SLRL2 Δ C3), which lacks the ability to interact with NF-YB1, was added and the NF-YB1 protein became more stable and was still present in the lane after 60 min of incubation (the eighth row).

All these data showed that SLRL2 competes with NF-YC12 to interact with NF-YB1, resulting in the formation of an NF-YB1-YC12 complex or an NF-YB1-SLRL2 complex, depending on the protein abundance of SLRL2 and NF-YC12. Interestingly, these two regulatory complexes played opposite roles in regulating *Wx* expression and

rice AC. To further test this hypothesis with in vivo data, we analyzed the expression of *NF-YC12* and *SLRL2* in the developing seeds with or without ABA treatment. The result showed that the expression of *NF-YC12* increased only slightly, while that of *SLRL2* was remarkably promoted in response to ABA treatment (Response Figure 2-2f), suggesting that less NF-YB1-YC12-bHLH144 complex is formed in the high ABA condition.

Response Figure 2-2. SLRL2 competes with NF-YC12 to interact with NF-YB1. a Diagrams for NF-YB1 protein truncations and functional motif. **b** Yeast two-hybrid assay for interactions between NF-YC12 and NF-YB1 truncations. **c** Yeast two-hybrid assay for interactions between SLRL2 and NF-YB1 truncations. **d** SLCA shows the inhibitory effect of SLRL2 on the interaction between NF-YC12 and NF-YB1. **e** Cell-free degradation analysis of NF-YB1-MYC co-incubated with different protein sets, including SLRL2, NF-YC12, bHLH144, SLRL2 Δ C3 and their different combinations. Protein samples were collected at the indicated time points for Western blot analysis. The amount of NF-YB1-Myc protein was detected using an anti-Myc antibody. **f** Expression analysis of NF-YC12, SLRL2, and NF-YB1 in seeds (15 DAF) in response

to ABA treatment. OsActin01 was used as an internal control to normalize the gene expression. The expression level of NF-YC12 with mock treatment was set to 1.

Wx gene is an important target in this study, so expression of Wx gene and Western blot of GBSS should be provided in the *slrl2*, *SLRL2-OX*, *nf-yb1* and *bhlh144* materials.

Response: Yes, following your suggestion, we have provided the data of *Wx* gene expression and Western blot of Wx protein in the materials of *slrl2*, *SLRL2-OX*, *nf-yb1* and *bhlh144* (Response Figure 2-3a-d). The quantification data of the RT-qPCR result of the *Wx* gene is accurate and clear. The expression of *Wx* was increased in the *slrl2* mutants and decreased in other samples tested, including *SLRL2-OX*, *nf-yb1* and *bhlh144* rice. As for the difference in Wx protein in different samples, the general trend is similar to that of the RT-qPCR result. However, it is not as distinct as the RT-qPCR result due to the slight difference between samples and the limitations of WB, a semi-quantification method. Thank you very much.

Response Figure 2-3. Expression analysis of *Wx* genes in *slrl2*, *SLRL2*

overexpression, *nf-yb1* and *bhlh144* mutants. a *Wx* gene expression levels in *SLRL2*-related materials. *OsActin01* was used as an internal control to normalize gene expression. The transcript abundance of *Wx* in the WT was set to 1. Error bars represent the SD (n = 3 biological replicates). **b** Analysis of *Wx* protein abundance in *SLRL2*-related rice materials. **c** Transcriptional analysis of the *Wx* gene in *nf-yb1* and *bhlh144* mutants. Error bars represent SD (n = 3 biological replicates). Different lowercase letters above the error bars indicate significant differences according to a one-way ANOVA with Tukey's test (P < 0.05). **d** Western blot analysis of *Wx* protein abundance in *nf-yb1* and *bhlh144* mutants.

In Fig. 6: the model did not show that the *SLRL2* works downstream of *NF-YB1* and *bHLH144*.

Response: We thank the reviewer expert for the critical comments. We have further improved the model based on your suggestions and the newly added data (Response Figure 2-4). We believe that the model is now more accurate and informative.

Response Figure 2-4. A model for the *NF-YB1*-*SLRL2*-*bHLH144* regulatory cascade in rice AC and PHS. At stages of rice seed development or under

environmental stress, ABA levels are elevated in the seed, which induces the expression of *SLRL2*. *SLRL2* is a DELLA family member with both transcriptional activation activity and direct DNA binding activity. Specifically, *SLRL2* is a negative regulator of rice AC by repressing the expression of the *Wx* gene, which is the primary key determinant of rice AC. *SLRL2* fine-tunes *Wx* gene expression through a variety of approaches. First, *SLRL2* binds directly to the *Wx* promoter and represses its expression. Second, *SLRL2* physically interacts with and represses the transcriptional activity of NF-YB1, a key positive regulator of *Wx* gene expression. Third, *SLRL2* binds directly to the *bHLH144* promoter and represses its expression. It is known that bHLH144 can increase the stability of NF-YB1 by forming a heterotrimer complex with NF-YB1 and NF-YC12. Fourth, *SLRL2* interacts with NF-YB1 to form a heterodimer, which is also helpful in increasing NF-YB1 stability. Finally, *SLRL2* competes with NF-YC12 to interact with NF-YB1, thereby affecting the formation of the NF-YB1-YC12-bHLH144 heterotrimer and decreasing NF-YB1 stability. Thus, a *SLRL2*-NF-YB1-bHLH144 transcriptional regulatory module is established for the fine regulation of rice AC. In addition, this molecular work module is also suitable for the regulation of rice PHS. *SLRL2* binds directly to the *MFT2* promoter and activates its expression. *MFT2* is a positive regulator of ABA signaling and promotes seed dormancy. Therefore, the *SLRL2* centered-regulatory module may also mediate ABA-regulated seed dormancy. Arrows and bars indicate activation and repression effects, respectively. Orange color indicates post-translational regulation. Blue color indicates transcriptional regulation. Solid and dashed lines indicate direct and indirect (or unknown mechanism) effects, respectively.

L263-264: this sentence is difficult to follow, what is the two sequences? In L272, it indicated that bHLH144 was included in the DAP-seq data, whether the two sequences including the bHLH144.

Response: We apologize for the unclear description. Here, a prey DNA library consisting of short random DNA sequences was used to identify the potential direct binding sequence of *SLRL2*. Each short random DNA sequence was ligated into the AD vector. Thus, all these random DNA-AD plasmids formed a prey DNA library. The *SLRL2*-BD and the prey DNA library were co-transformed into yeast and the clones grown up in the selection medium were sequenced. The frequency of the sequences that appeared gave us some clues about the *SLRL2* binding site. Here, the sequence

“CCCTCCATCAAACGGG” had the highest frequency of occurrence, accounting for 56.4% of the total sequenced clones. And the sequence “CCCAAGAGGCGATGGG” was the second most frequent, accounting for 30.8% of the total sequenced clones. These two sequences were shown in Response Figure 2-5a (the first two rows). By analyzing these two sequences, both of them contained “CTCC/GAGG”. Therefore, we proposed that this sequence must be conserved in the binding motif of SLRL2. Subsequently, DAP-seq analysis identified the conserved sequence of the SLRL2 binding site as “CTCCCTCCGT”, which also contained the key sequence “CTCC” (Response Figure 2-5b). We have improved the description in the revised manuscript.

In the original L272, the conserved key sequence “CTCC” was also contained in the *bHLH144* promoter (Response Figure 2-5c). Thank you very much.

Response Figure 2-5. Analysis of transiently acting components of SLRL2 binding. **a** Yeast one-hybrid library screening to identify the potential SLRL2 binding site. Percentages represent the frequency of occurrence of the cis-acting element. **b** Conserved DNA binding site of SLRL2 identified by DAP sequencing. **c** Schematic representation of the *bHLH144* promoter and the potential SLRL2 binding site.

L53: What is soft rice, whether is it a scientific term that has been accepted in rice community.

Response: Thank you for your critical comments. Soft rice refers to high quality rice

with superior ECQ, characterized by a soft and elastic texture, and relatively low amylose content (AC), generally around 10% to 14%. Soft rice is becoming increasingly popular with consumers, especially those in the Yangtze River Delta region of China. Actually, soft rice is a relatively new term and has started to be accepted in the rice community in recent years (Li et al., 2018; Zhang et al., 2019; Liu et al., 2019; Wang et al., 2020; Zhang et al., 2021; Huang et al., 2021; Yang et al., 2022; Fan et al., 2023; Kalita et al., 2023; Zhang et al., 2023), especially accompanied with the continuous improvement of people's quality of life and the increasing demand for high-quality rice. Perhaps soft rice is not a scientific term, but it is an appropriate choice to indicate this specific type of high quality rice at present.

References:

- Li, Q. F. et al. Down-Regulation of *SSSII-2* Gene Expression Results in Novel Low-Amylose Rice with Soft, Transparent Grains. *J Agric Food Chem.* **66**(37), 9750–9760 (2018)
- Zhang, H. et al. The *qSAC3* locus from indica rice effectively increases amylose content under a variety of conditions. *BMC Plant Biol.* **19**(1), 275 (2019)
- Liu, Y. et al. Development and validation of a PCR-based functional marker system for identifying the low amylose content-associated gene *Wx^{hp}* in rice. *Breed Sci.* **69**(4), 702–706 (2019)
- Wang, L., Gong, Y., Li, Y., & Tian, Y. Structure and properties of soft rice starch. *Int J Biol Macromol.* **157**, 10–16 (2020)
- Zhang, C. et al. A rare *Waxy* allele coordinately improves rice eating and cooking quality and grain transparency. *J Integr Plant Biol.* **63**(5), 889–901 (2021)
- Huang, X. et al. Novel *Wx* alleles generated by base editing for improvement of rice grain quality. *J Integr Plant Biol.* **63**(9), 1632–1638 (2021)
- Yang, J. et al. Development of Soft Rice Lines by Regulating Amylose Content via Editing the 5'UTR of the *Wx* Gene. *Int J Mol Sci.* **23**(18), 10517 (2022)
- Fan, P. et al. Phenotypic differences in the appearance of soft rice and its endosperm structural basis. *Front Plant Sci.* **14**, 1074148 (2023)

Kalita, P., Ahmed, A. B., Sen, S., Pachuau, L., & Phukan, M. Synthesis and characterization of citrate soft rice starch: A new strategy of producing disintegrating agent for design drug and resistant starch. *Int J Biol Macromol.* **240**, 124475 (2023)

Zhang, C., Xue, W., Li, T., & Wang, L. Understanding the Relationship between the Molecular Structure and Physicochemical Properties of Soft Rice Starch. *Foods.* **12**(19), 3611 (2023).

Thank you again for all your critical comments and constructive suggestions.

REVIEWERS' COMMENTS

Reviewer #1 (Remarks to the Author):

The author has addressed the most of concerns from me and other reviewers, and substantially improved the MS especially the experiments. But I still have one question about the CHIP experiment in your MS, for the seed trait or seed specific gene, we should perform CHIP analysis with seeds material instead of leaf.

Reviewer #2 (Remarks to the Author):

Authors have addressed my concerns and have amended the manuscript appropriately. I agreed the novel findings pointed out by authors, and I have no further comments.

Response to Reviewers' comments

REVIEWER COMMENTS

Reviewer #1 (Remarks to the Author):

The author has addressed the most of concerns from me and other reviewers, and substantially improved the MS especially the experiments. But I still have one question about the CHIP experiment in your MS, for the seed trait or seed specific gene, we should perform CHIP analysis with seeds material instead of leaf.

Response: Thank you very much for your positive comments and valuable suggestions. Yes, we agree with you that it's better to perform ChIP analysis with seeds material instead of leaf. Therefore, 7 day-after-flowering (DAF) developing seeds of *SLRL2-Flag* transgenic rice were used for ChIP analysis. The results of ChIP-qPCR are consistent with previous results obtained from rice leaf samples. The data indicated that NF-YB1 has the capability to bind to the promoters of *Wx* and *SLRL2*, respectively (Response Figure 1 a, b). While *SLRL2* has the capability to bind to the promoters of *Wx* and *bHLH144*, respectively (Response Figure 1 c, d). We have replaced the original ChIP-qPCR data with the new data.

Response Figure 1. ChIP-qPCR assay using rice seed material. a ChIP-qPCR assay

showed that the P1 region of the *Wx* promoter was significantly enriched for the transcription factor NF-YB1. **b** The ChIP-qPCR assay showing the enrichment of the *SLRL2* promoter by NF-YB1 in rice seeds. **c** ChIP-qPCR assay showed that the P1 region of the *Wx* promoter was also significantly enriched by SLRL2. **d** ChIP-qPCR assay showed that the P1 region of the *bHLH144* promoter was significantly enriched by SLRL2. Enriched values are normalized to the input. An unrelated *OsActin1* intron region was used as a negative control. IgG-immunoprecipitated DNA was used as a control. Data are means \pm SD (n = 3 independent experiments). Statistical analysis was performed by two-tailed Student's *t*-test.

Reviewer #2 (Remarks to the Author):

Authors have addressed my concerns and have amended the manuscript appropriately. I agreed the novel findings pointed out by authors, and I have no further comments.

Response: We are pleased that our revisions have addressed all of your comments and questions to your satisfaction. Thank you very much.